# Characterization of genome-wide STR variation in 6487 human genomes

Yirong Shi[1,2,6], Yiwei Niu [1,3,6], Peng Zhang[1], Huaxia Luo[1], Shuai Liu[1,3], Sijia Zhang[1,3], Jiajia Wang[1], Yanyan Li[1], Xinyue Liu[1,2], Tingrui Song[1], Tao Xu [4,5] ✉ & Shunmin He [1,3] ✉

Short tandem repeats (STRs) are abundant and highly mutagenic in the human genome. Many STR loci have been associated with a range of human genetic disorders. However, most population-scale studies on STR variation in humans have focused on European ancestry cohorts or are limited by sequencing depth. Here, we depicted a comprehensive map of 366,013 polymorphic STRs (pSTRs) constructed from 6487 deeply sequenced genomes, comprising 3983 Chinese samples (~31.5x, NyuWa) and 2504 samples from the 1000 Genomes Project (~33.3x, 1KGP). We found that STR mutations were affected by motif length, chromosome context and epigenetic features. We identified 3273 and 1117 pSTRs whose repeat numbers were associated with gene expression and 3′ UTR alternative polyadenylation, respectively. We also implemented population analysis, investigated population differentiated signatures, and genotyped 60 known disease-causing STRs. Overall, this study further extends the scale of STR variation in humans and propels our understanding of the semantics of STRs.

Short tandem repeats (STRs; also known as microsatellites) are 1–6 base pair (bp) tandem repeats, accounting for approximately 3% of the human genome[1,2]. The repetitive structure endows STRs with a higher mutation rate than other parts of the genome[3,4], ranging from approximately $10^{-5}$ to $10^{-3}$ per locus per generation[5]. Most mutations of STRs are due not to substitutions but rather to expansions or contractions of repeat units[6], resulting in digital length polymorphisms[3]. After the seminal discovery that expansions of CGG repeats in the *FMR1* gene were linked to fragile X syndrome (FXS) in 1991[7–10], researchers have identified approximately 60 STR loci implicated in a range of Mendelian diseases to date, including ataxias, amyotrophic lateral sclerosis, Huntington disease, frontotemporal dementia, and various neurological disorders[11,12]. More importantly, although our view remains incomplete, emerging evidence has shown that a significant number of polymorphic STRs (pSTRs) can modulate various

molecular and cellular processes, such as DNA methylation[13], gene expression[13–18], and alternative splicing[19–22], suggesting that pSTRs may contribute to complex phenotypes[3,23].

The last decade has witnessed the great success of genome-wide association studies (GWASs) in human genetics, reporting tens of thousands of single nucleotide polymorphisms (SNPs) associated with over 1000 traits[24]. However, one major concern of most GWASs on complex traits and polygenic disorders is the "missing heritability" problem[25], partly due to the standard GWAS analysis focused only on SNPs[26]. Although overlooked by most GWASs and other association studies, it is widely believed that pSTRs may contribute to missing heritability of human traits and disorders[3,12,23,27,28]. Recently, a phenome-wide association study identified 426 tandem repeat-phenotype associations, in which GT repeats in *NCOA6* and "ease of skin tanning" were the most significant[29]. Furthermore, Margoliash

[1]Key Laboratory of RNA Biology, Center for Big Data Research in Health, Institute of Biophysics, Chinese Academy of Sciences, Beijing 100101, China. [2]University of Chinese Academy of Sciences, Beijing 100049, China. [3]College of Life Sciences, University of Chinese Academy of Sciences, Beijing 100049, China. [4]National Laboratory of Biomacromolecules, CAS Center for Excellence in Biomacromolecules, Institute of Biophysics, Chinese Academy of Sciences, Beijing 100101, China. [5]Shandong First Medical University & Shandong Academy of Medical Sciences, Jinan 250117 Shandong, China. [6]These authors contributed equally: Yirong Shi, Yiwei Niu. ✉e-mail: xutao@ibp.ac.cn; heshunmin@ibp.ac.cn

et al. tested the association between STRs and blood and serum traits using imputed STR genotypes and identified 118 high-confidence STR-trait associations[30]. These studies greatly inspired us to discover trait/disease-associating pSTRs; unveiling the contributions of pSTRs in a range of complex traits would be an important long-term goal[12,31]. To achieve this, one urgent need is to construct a full and accurate catalog of pSTRs in the human genome[3]. Such a resource would also propel our understanding of the mutational patterns and functional impacts of pSTRs and serve as a reference point for identifying STR variants in disease contexts[28].

With advances in DNA sequencing technologies and computational approaches for STR genotyping[12], there has been significant progress in profiling STR polymorphisms in diverse control populations[32–39]. However, these studies were limited by sample number, population diversity, sequencing depth, or algorithms employed; genome-wide pSTR data in diverse populations are still insufficient[3]. For example, the largest genome-wide study of STR variation to date genotyped 2,536,688 STR loci in 150,119 genomes from the UK biobank using popSTR[38]. However, most individuals in this study were white British, and popSTR is only able to genotype STR alleles shorter than the read length[40]. Moreover, other large-scale studies of pSTRs were also mainly from European ancestry cohorts, and the diversity of pSTRs in East Asia and China is largely under-covered. Even in the UK biobank cohort, there were only 1504 Chinese samples[38]. As the Han Chinese population is the largest ethnic group in East Asia and in the world[41], constructing a comprehensive map of pSTRs from the Han Chinese population is imperative and would help to solve the critical part of the missing diversity.

Here, we collected whole genome sequencing (WGS) data of 3983 samples from the NyuWa dataset and 2504 samples from the high-coverage 1KGP dataset to identify pSTRs. The samples in the NyuWa dataset were mostly Han Chinese, in which 2999 members were previously used for investigating small variants[42] and mobile element insertions[43]. The deeply sequenced 1KGP dataset[44] has been used to detect de novo tandem repeat expansions[45] and reference minisatellite variable number tandem repeats (VNTRs)[46,47], while we included it here to study variations of reference STRs and to increase population diversity. Jointly analyzing two large cohorts enabled us to obtain a systematic view of the STR variation in diverse populations, with an emphasis on Han Chinese. GangSTR[48] was employed to collect information in the 765,227 autosomal STR loci, as it is the only algorithm capable of accurately genotyping both short and expanded STRs[40,49,50]. We first surveyed the pSTR call set for their basic characteristics, such as allele frequency, genetic diversity, genomic distributions, and functional consequences. We identified pSTRs with potentially functional impact through loss-of-function analysis, linkage disequilibrium (LD) with GWAS SNPs, and expansion analysis. By utilizing public RNA-seq data, we identified pSTRs affecting nearby gene expression and 3′UTR alternative polyadenylation. We then performed population analysis and identified many pSTRs with significant length differences between and within superpopulations. Finally, we genotyped 60 known disease STRs in all samples with ExpansionHunter[51] and provided the population-wide allele distributions of these loci. This study represents one of the largest and latest genome-wide studies of STR variation in various populations and will further our understanding of how this mutagenesis impacts the human genome.

## Results

### The pSTR call set
We generated a genome-wide map of STR variation by jointly analyzing two cohorts: the NyuWa dataset consisting of 3983 Chinese individuals sequenced to ~31.5X coverage[42] and the 1KGP dataset consisting of 2504 samples sequenced to ~33.3X coverage (Supplementary Data 1)[44,52]. We applied GangSTR[48] to genotype 765,227 autosomal STRs with repeat unit lengths of 2–6 bp in each genome. With good

efficiency and scalability, GangSTR is capable of genotyping both contracted and expanded STRs by leveraging evidence beyond repeat-enclosing reads[49,50] and has been utilized in other large-scale studies[29,53]. To investigate pathogenic STRs and supplement our call set, we also collected information on 60 known disease-causing STRs (Supplementary Data 2; Fig. S1) in our cohorts by using another STR genotyper, ExpansionHunter[51]. Similar to GangSTR, ExpansionHunter outputs maximum-likelihood genotypes consisting of candidate repeat alleles by incorporating multiple features of paired-end reads. For the seven sites genotyped by both GangSTR and ExpansionHunter, their estimated repeat lengths were highly concordant (median concordance rate: 99.2%; Supplementary Data 3).

To ensure the call set quality of GangSTR, we removed genotypes with read coverage that was too low or too high, with reads supported by only spanning or flanking reads, and with maximum likelihood repeat lengths falling outside 95% confidence intervals. After call-level filtering, the average call rates of samples in the NyuWa dataset and 1KGP dataset were 95.9% (Fig. S2a) and 98.4% (Fig. S2b), respectively. Additional site-level filters, including STRs in segmental duplications, STRs with call rates <20%, or genotypes violating Hardy-Weinberg equilibrium, were also applied to further improve the call quality. These filtering steps increased the accuracy of our calls, as indicated by the improvement of Mendelian inheritance rates of STRs (Fig. S2c). An average call rate of 98.3% was obtained for all STR loci (Fig. S2d), and call rates did not show a marked decrease when the reference allele lengths increased (Fig. S2e). For pSTRs with reference allele lengths ≤40 bp, which accounted for 98.2% of all pSTRs, the mean differences in length of each allele called compared to the reference allele were approximately zero (Fig. S2f); the genotypes had high-quality scores at both the sample level and site level (Fig. S2g, h). These data collectively indicated that our catalog represents a high-quality map of STR variation for humans.

In total, our analysis identified 366,013 pSTRs in the 6487 WGS samples, with 306,602 and 276,294 pSTRs detected in the NyuWa dataset and 1KGP dataset, respectively (Fig. 1a). Comparing pSTRs detected from the NyuWa and 1KGP datasets, 89,719 pSTRs and 523,063 alleles were specific to the NyuWa dataset (Fig. 1b), with more low-frequency alleles detected in the NyuWa dataset (Fig. S3a, b). When restricted to East Asian individuals in the 1KGP dataset, 93.4% pSTRs and 92.6% alleles could be detected in the NyuWa samples, with more sites and alleles specific to the NyuWa samples, indicating the great value of the NyuWa dataset in profiling STR variation (Fig. 1b). We confirmed that the dataset-specific loci were not due to low call rates in another dataset (Fig. S3c, d). The major allele frequencies of shared pSTRs between the NyuWa dataset and 1KGP dataset showed high correlation, especially for pSTRs of EAS in 1KGP (Fig. S3e, f). Next, we estimated the total pSTRs and the increase in pSTRs at different sample sizes by randomly downsampling to different sizes with 100-sample intervals. While the number of total pSTRs continued to rise with increasing sample size, the number of common pSTRs plateaued at a sample size of 100 (Fig. S4), consistent with a previous study[33].

For controls, we also identified 290,454 high-quality monomorphic STRs (mSTRs) in all samples analyzed using the same stringent filters (see "Methods"). For the reference STR set we analyzed, there were more pSTRs than mSTRs for di- ($\chi^2$ $P < 1 \times 10^{-22}$), tri- ($\chi^2$ $P < 1 \times 10^{-22}$), and tetranucleotide ($\chi^2$ $P = 8 \times 10^{-22}$) STRs (Fig. S5a, b), implying that STRs with shorter repeat units had higher plasticity, as previously reported[33,54]. In addition, the reference allele lengths of pSTRs were significantly longer than those of mSTRs (Fig. S5c; median of pSTRs: 15, median of mSTRs: 12), suggesting an increased mutation rate in longer reference allele lengths[3,33].

### pSTR mutational patterns
With the comprehensive repertoire of pSTRs, we next examined the mutational patterns of STRs. We found that over half of the pSTRs

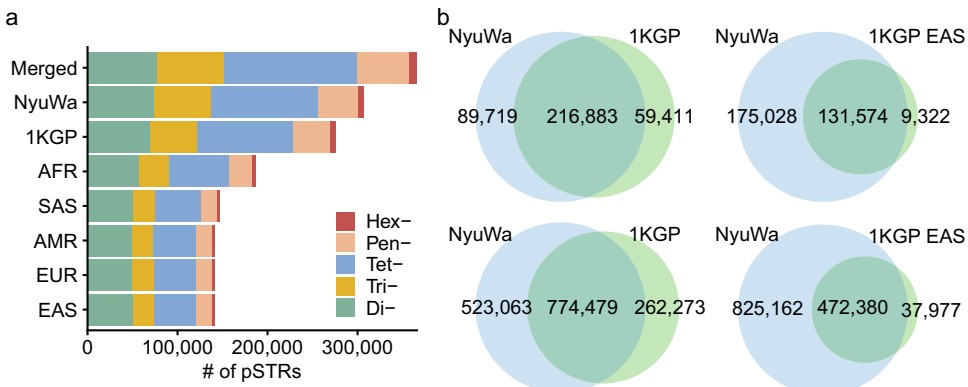

**Fig. 1 | pSTR identified in this study. a** The cumulative number of pSTR loci broken down by dataset, superpopulation, and STR types. **b** Comparison of the pSTR loci (upper) and pSTR alleles (lower) identified from the NyuWa dataset with those identified from the 1KGP dataset (left) and the East Asian samples in the 1KGP dataset (right).

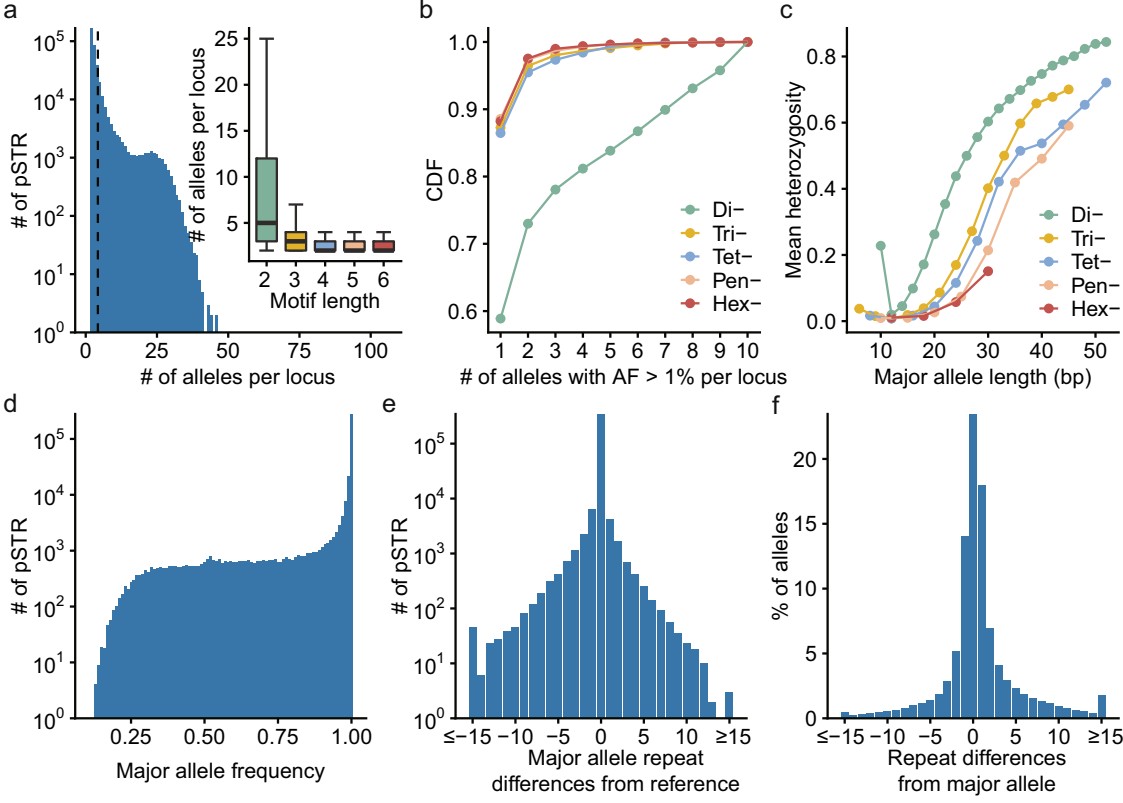

**Fig. 2 | pSTR mutational patterns. a** Distribution of the number of alleles per pSTR locus ($n = 366,013$). The black dashed line indicates the mean value (4.26) of allele numbers of pSTRs. The inner boxplot shows the same variable grouped by motif length. Horizontal lines indicate the median and boxes span from the lower quartile (the 25th percentiles) to the upper quartile (the 75th percentiles). Whiskers extend to points that are within 1.5× IQR (interquartile range) from the upper or the lower quartiles. **b** Cumulative distribution function (CDF) of the number of alleles with frequency >0.1% per pSTR locus, which was classified by motif length. **c** pSTR heterozygosity as a function of the length of the major (most common) allele in base pairs, which was classified by motif length. **d** Distribution of the frequency of the major allele per pSTR locus. **e** Distribution of the differences in the repeat number of the major allele from the reference allele per pSTR locus. **f** Distribution of the difference in the repeat number of each pSTR allele from the major allele of the corresponding pSTR locus.

(54.2%) were highly multiallelic (mean = 4.26 alleles, median = 3 alleles), and the allele number per locus decreased with motif length (mean: 8.82 to 2.61) (Fig. 2a), consistent with previous observations in humans[33], pigs[55], *C. elegans*[56], and *A. thaliana*[57]. The maximum allele numbers of di-, tri-, tetra-, penta-, and hexanucleotide pSTRs were 51, 38, 73, 105, and 15, respectively. When restricted to common alleles (allele frequency > 1%), 18.8% of pSTRs had at least two common alleles, and dimeric pSTRs had the highest proportion of loci with multiple common alleles (Fig. 2b). These data indicated that STRs with shorter

motif lengths showed a higher degree of length polymorphism; this still held true when stratifying STRs by major allele length (Fig. 2c), consistent with a previous report[33].

Similar to the observations in *C. elegans*[56], only 3.6% of loci had a major allele frequency less than 0.5 (Fig. 2d; Fig. S6a), likely because of the loss of genetic diversity resulting from recent explosive human population growth[58]. Meanwhile, repeat distances between the most common alleles and reference alleles revealed a symmetric spectrum, with the reference alleles of the vast majority (94.6%) of pSTRs being

the most common (Fig. 2e; Fig. S6b), largely consistent with a previous report[33]. For dinucleotide STRs, the major alleles of 16.2% (12,559/77,617) of the loci had at least one repeat away from the reference alleles, reflecting the high instability of these pSTRs (Fig. S6b). For each pSTR, we then compared the repeat number of reported alleles with the most common allele (Fig. 2f). We observed a higher proportion of alleles longer than the major allele (45.2%) compared to alleles shorter than the major allele (31.3%), and the allele frequency monotonically decreased with deviation from the major allele (Fig. 2f; Fig. S6c), coinciding with the stepwise mutation model[59]. We also performed this analysis per motif length and per major allele length, and similar trends were observed (Fig. S7).

We next investigated the chromosome distribution of pSTRs, as STRs were nonuniformly distributed in the genome[2]. We observed a modest enrichment within subtelomeric regions for hexameric pSTRs, while no such bias was found for other types of pSTRs or mSTRs (Fig. S8). Compared with other regions on the chromosome, there was a significant increase (1.6-fold increase for the p arms and 1.8-fold increase for the q arms) in hexameric pSTR density in the last 5 Mbp of chromosome arms (Permutation $P < 1 \times 10^{-4}$). We also applied another method for investigating the chromosome distribution of structural variants (SVs) to validate this result[60]. As expected, we also found significant enrichment of hexanucleotide pSTRs in subtelomeric regions (Fig. S9; Fig. S10). For comparison, a similar analysis was applied to mSTRs, but no such patterns for hexameric mSTRs were found (Fig. S11). Previous studies have found a strong enrichment of VNTRs in subtelomeres[32,61,62] and associated it with double-strand breaks[62,63]. We also observed a modest correlation between hexameric pSTRs and double-strand breaks defined by PRDM9 protein binding hotspots in DNA[63] (Fig. S12). These analyses implied the influence of chromatin context and STR motif length in determining STR mutation rates throughout the genome.

As genomic niches may influence STR instabilities[3,5], we next correlated the occurrences of pSTRs and mSTRs with various genome features (Fig. S13; Supplementary Data 4). Largely in line with previous reports, we found that pSTRs were positively correlated with gene count[5], GC percent, active transcription[64], and various euchromatin markers, including H3K9ac, H3K27ac, and H3K4me3. Instead, pSTRs were negatively correlated with markers of heterochromatin, such as CpG DNA methylation and H3K9me3 (Fig. S13a, c), a reflection of the links between epigenetic changes and STR instability[3]. Of note, we observed that dinucleotide pSTRs occurred less often in transcriptionally active regions, possibly due to the depletion of dinucleotide STRs in genic regions[65]. In contrast, we did not observe such patterns for mSTRs (Fig. S13b, d).

## pSTR functional properties

Fluctuations in the length of pSTRs in specific genic and intergenic regions can result in a variety of biological consequences via different molecular mechanisms[3,12]. To assess the functional properties of pSTRs, we first annotated pSTRs using Variant Effect Predictor (VEP). We found that over ninety percent of pSTRs were in intronic (58.5%) and intergenic (34.5%) regions, and only 1881 pSTRs were within coding sequences (CDSs), of which 1426 (75.8%) and 256 (13.6%) loci were trimeric and hexameric pSTRs, respectively (Fig. 3a). We then examined the enrichment of STRs in different genic features by applying a simulation-based method. Varying enrichment levels in different genomic regions were observed for both pSTRs (Fig. 3b) and mSTRs (Fig. 3c). In line with a previous report[66], both pSTRs and mSTRs were generally enriched in 5′UTRs and depleted in CDSs, and STRs with different motif lengths showed dissimilar distributions across genomic regions (Fig. 3b, c). Compared with their mSTR counterparts, di-, tetra-, and pentanucleotide pSTRs were more highly depleted in coding regions (Fig. 3b, c), reflecting their deleterious effects on gene function[67]. In contrast, tri- and hexameric STRs were overrepresented

in 5′UTRs and CDSs compared with other STR types. This was likely because triplet STRs are all in-frame indels that should have less of an effect on transcript and protein function than other nontriplet STRs[6,68]. Gene Ontology analyses showed that genes containing trimeric and hexameric pSTRs in the 5′UTR and coding regions were enriched in gene categories related to development and differentiation (Fig. S14), in line with a previous report[69].

To examine the selective constraint of pSTRs, we compared the heterozygosity and entropy of each pSTR in different genomic regions (Fig. 3d, e; Fig. S15). Using pSTRs in coding gene introns as controls, pSTRs in coding regions showed significantly lower heterozygosity and entropy, as previously reported[6]. In coding regions, pSTRs with motifs of multiples of three nucleotides had higher variability than other pSTR types (Fig. 3d, e; Fig. S15). From the perspective of pSTR-containing genes, we utilized another measure: the loss-of-function observed/expected upper bound fraction (LOEUF), where higher LOEUF scores indicate higher tolerances to inactivation for given genes[70]. Higher LOEUF scores were observed for genes with pSTRs in coding regions than those with pSTRs in intronic regions (Fig. 3f).

In principle, pSTRs residing in coding regions can result in loss-of-function (LoF) by disrupting open-reading frames or transcript splicing. We identified 668 LoF alleles at 392 pSTR loci, including 254 frameshift mutations and 394 splicing region variants (Fig. S16a, b; Supplementary Data 5). Most LoF alleles were low-frequency (allele frequency <0.0001), especially frameshift variants (Fig. S16b), indicating their harmful impacts on normal gene function. Consistently, the genetic plasticity of LoF pSTRs was significantly lower than that of other pSTR loci (Fig. S16c), a reflection of purifying selection against LoF variants.

It has long been hypothesized that tandem repeat variation contributes to complex traits in humans[3,12,23,27,28]. To identify pSTRs potentially associated with complex human traits or diseases, we calculated the LD in terms of the $r^2$ between pSTRs and trait- or disease-associated loci identified by GWASs ($P \leq 5 \times 10^{-8}$)[24]. We identified 2871 pSTRs that were in high LD ("tagged") with at least one GWAS SNP ($r^2 \geq 0.7$), accounting for approximately 0.78% of all pSTRs (Fig. S17a; Supplementary Data 6). The major allele frequencies for 303 of these pSTRs were less than 0.5. Traits with the highest numbers of GWAS-tagged pSTRs included height, body mass index, and mineral density of the heel bone (Fig. S17b). These data exemplified the value of pSTRs in complex phenotypes and the prospects of our pSTR call set for future genotype-phenotype association studies.

## pSTR gene-regulatory effects

pSTRs can affect variable phenotypes and disease susceptibility through their gene-regulatory effects[3,12]. Previous genome-wide searches have identified thousands of pSTRs associated with human gene expression[13–18], but our understanding of this landscape remains incomplete. Moreover, the direct exploration of pSTRs associated with gene posttranscriptional regulation has not been performed, although several single-gene studies have reported that pSTRs can modulate RNA structure and function[3]. Here, we aimed to identify pSTRs associated with total gene expression level (eSTRs) and 3′UTR alternative polyadenylation (3′aSTRs) by leveraging public RNA-seq datasets of lymphoblastoid cell lines (LCLs) from the Geuvadis project[71]. The LCL dataset contained 445 individuals of European and African ancestry overlapping with the 1KGP dataset, and some samples of this dataset have been used for the identification of eSTR associations[13,15]. After filtering lowly expressed genes and controlling for confounders (Fig. S18), we paired pSTRs with genes within 500 kb and applied a linear model to detect eSTR and 3′aSTR associations. For eSTR associations, a total of 293,991 gene-STR pairs were tested, and 4131 pairs (4131 genes, 3273 pSTRs) exhibited significant associations at a gene-level false discovery rate (FDR) of 10% (Supplementary Data 7; Fig. S19). The

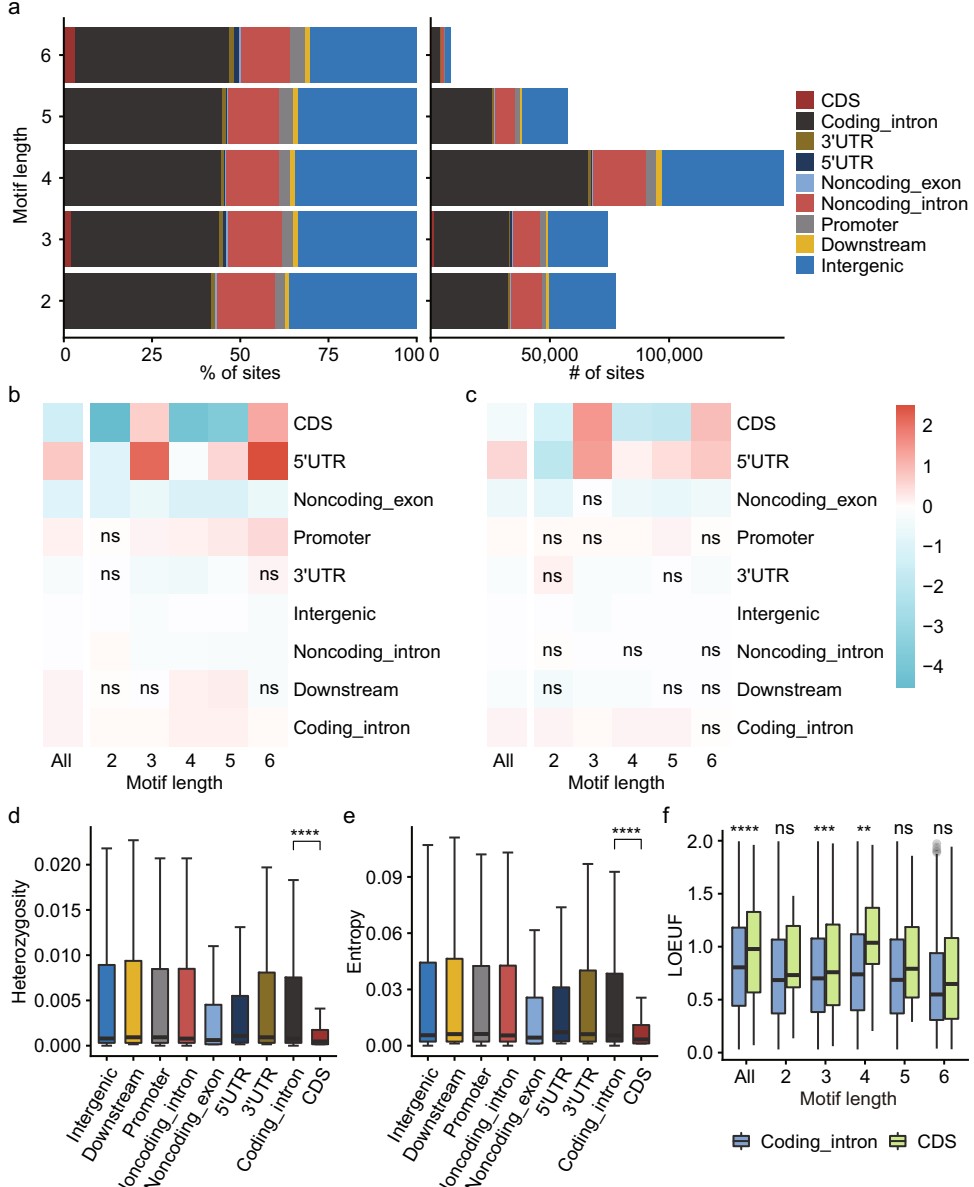

**Fig. 3 | pSTR functional properties. a** Functional consequence for pSTRs stratified by motif length: (left) cumulative proportion and (right) cumulative number. Coding_intron, introns of protein-coding genes; Noncoding_exon, exons of noncoding genes; Noncoding_intron, introns of noncoding genes. **b** Log2-fold enrichment of the pSTR call set compared against the pSTRs permutated. **c** Log2-fold enrichment of the mSTR call set compared against the mSTRs permutated. For both Fig. **b** and **c**, a permutation test was repeated 1000 times, and empirical *P* values were computed together with the enrichment values by GAT v1.3.4 and adjusted using Benjamini–Hochberg method. ns, not significant (adjusted *P* value >0.05). **d, e** Heterozygosity (**d**) and entropy (**e**) of pSTR loci (*n* = 366,013) in different genomic regions. **f** LOEUF scores of protein-coding genes enclosing pSTRs in the CDS (*n* = 282) and introns (*n* = 11,349). For Figure **d**–**f**, horizontal lines indicate the median, boxes span from the lower quartile (the 25th percentiles) to the upper quartile (the 75th percentiles), and whiskers extend to points that are within 1.5 × IQR (interquartile range) from the upper or the lower quartiles; the two-sided Wilcoxon rank-sum test was used to compute *P* values. ns, *P* value ≥ 0.05; **P value < 0.01; ***P value < 0.001; ****P value < 0.0001.

associations were corroborated by permutation controls (Fig. 4a) and a previous study (Fig. 4b), which used 311 European individuals from the LCL dataset to detect eSTRs[15]. We also compared our eSTR associations with eSTRs detected using expression data from the Genotype–Tissue Expression Project[16] and found good consistency (Fig. S20), with differences observed in various tissues. Using a similar method, we tested 967,309 transcript-STR pairs to identify 3′aSTRs, and 1984 pairs (1984 transcripts, 1117 pSTRs) were significant at transcript-level FDR < 10% (Fig. 4c; Supplementary Data 8). For many pairs, clear relationships between repeat numbers and the percentage of distal poly(A) site usage index (PDUI) values of transcripts could be observed (Fig. S21).

We next sought to investigate the genomic contexts of eSTRs and 3′aSTRs to understand their functional mechanisms in gene regulation. We found that eSTRs as well as 3′aSTRs were more concentrated in regions with active histone marks (e.g., H3K9ac, H3k4me3, H3K4me2, and H3K27ac) and open chromatin (Figs. 4d, S22a), largely consistent with the observations of previous studies[15,16]. We found that eSTRs were most enriched in 5′UTR regions and depleted in CDSs, while 3′ aSTR were more frequently found in 3′UTR regions (Fig. 4d). Concordantly, eSTRs in 5′UTRs and 3′aSTRs in noncoding exons and 3′ UTRs had larger effect sizes than those in other regions (Fig. S23). We also performed the same enrichment analysis using the chromatin states defined by ChromHMM[72]. We found that both eSTRs and 3′

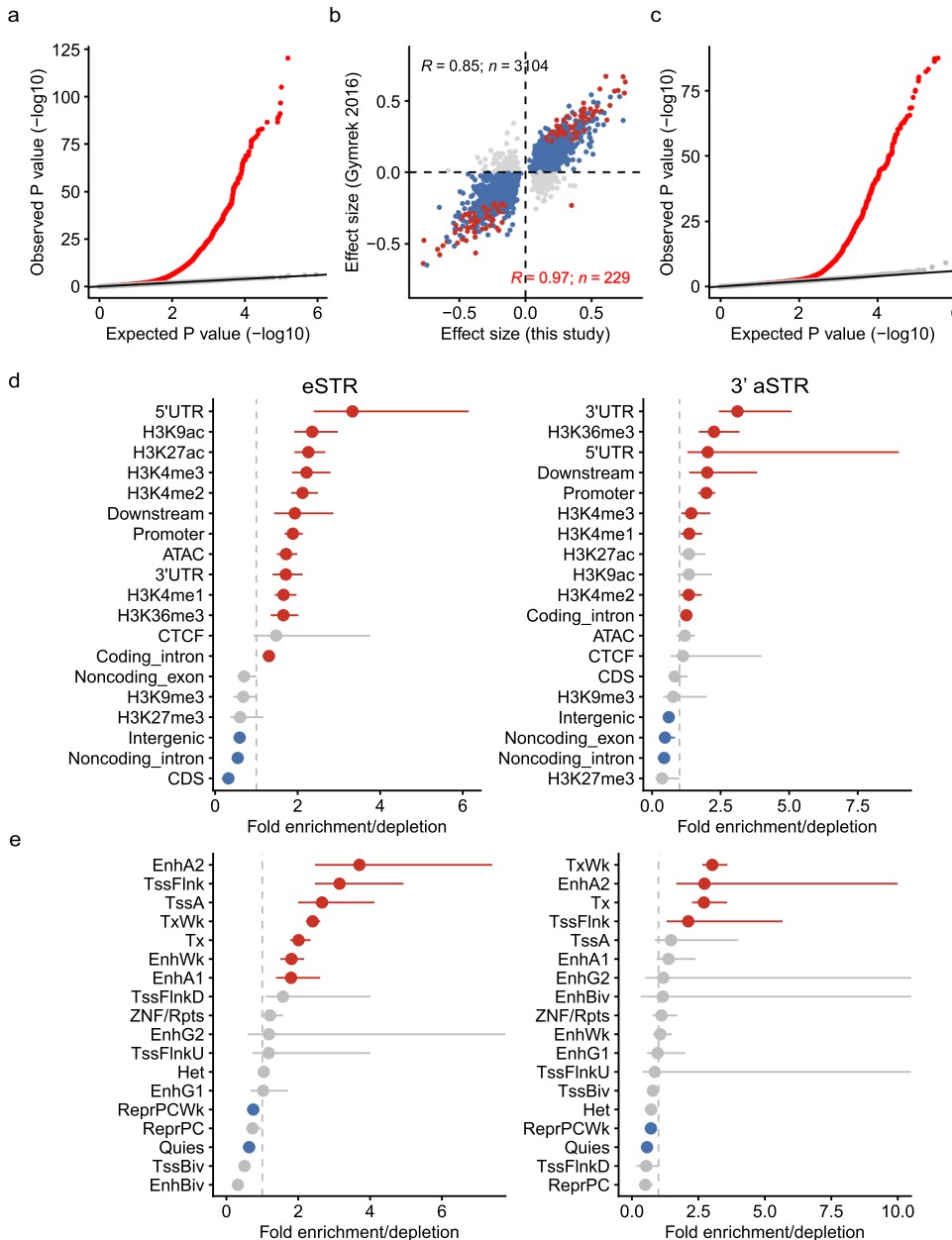

**Fig. 4 | eSTRs and 3'aSTRs identified in this study. a** Quantile–quantile plot comparing observed *P* values for STR-gene association tests (two-sided *t*-test in linear model) versus the expected uniform distribution in eSTR analysis. The red dots represent the observed association tests, and the gray dots indicate *P* values for permutation control. The black line gives the expected *P* value distribution under the null hypothesis of no association. **b** Correlations of the effect size of eSTRs identified in this study and a previous study by Gymrek et al. The blue points indicate eSTRs whose directions of effect were concordant in two studies, and gray points denote eSTRs with discordant directions of effect for that eSTR. The eSTRs detected in both studies are colored red, regardless of the concordance of effect.

**c** Quantile–quantile plot comparing observed *P* values for STR-gene association tests (two-sided *t*-test in linear model) versus the expected uniform distribution in 3'aSTR analysis. **d, e** Fold enrichment of eSTRs (left; *n* = 3273) or 3'aSTRs (right; *n* = 1117) in designated genome regions (**d**) and chromatin states defined by ChromHMM (**e**) in the GM12878 cell line. A permutation test was repeated 1000 times, and empirical *P* values were computed together with the enrichment values by GAT v1.3.4. Points denote the enrichment values. Red and blue points denote significant enrichments or depletions (*P* < 0.05 after Benjamini & Hochberg correction), and bars show 95% confidence intervals.

aSTRs were significantly concentrated in active enhancers and regions flanking transcription start sites (TSSs) (Figs. 4e, S22b, c). These results revealed that eSTRs and 3'aSTRs had generally similar genomic characteristics, and pSTRs in regulatory elements and accessible chromatin regions often exerted gene-regulatory effects.

To further investigate the potential impacts of eSTRs and 3'aSTRs on human traits, we compared the genomic locations of these pSTRs with GWAS signals. Relative to pSTRs included in eSTR/3'aSTR analyses, the observed number of both eSTRs and 3'aSTRs tagged by

GWAS SNPs was significantly greater than expected (*P* < 0.001, Fig. S24a, c), implying that these loci may have functional impacts found at multiple GWAS SNPs. We also tested whether eSTR- or 3'aSTR-associated genes were enriched in any gene sets implicated by previous GWAS findings, as previously described[15]. We observed that genes targeted by eSTRs were significantly enriched in GWAS genes for allergic rhinitis, primary biliary cirrhosis, eczema, etc. (Fig. S24b). For genes regulated by 3'aSTRs, we found that they were enriched in GWAS genes for primary biliary cirrhosis, sarcoidosis, myositis, etc. (Fig.

S24d). These results suggested that pSTR-associated gene modulation effects could introduce variability in complex traits and polygenic disorders. The most enriched GWAS traits we found here were relevant to the immune system, partly due to the cell type (lymphoblastoid cell lines) we used. It is also worth noting that our analyses in one cell type may be limited in finding associations with other complex traits[73].

## Population analysis of pSTR

STRs are powerful markers in population genetics studies[74]. While numerous studies have used STRs to examine variation patterns in diverse human populations[33,34,39,75,76], these studies were limited by locus number, sample size or population diversity. We first utilized our comprehensive catalog of pSTRs to study population structure using principal component analyses and found that the clusters generally resembled the trends found with SNPs[42] and SVs[77] (Fig. S25a). Principal component 1 (PC1) separated individuals of African ancestry from those of non-African ancestry, while American populations were scattered among Europeans, South Asians, and East Asians; PC3 separated populations of South Asians from all other individuals. We next performed the same analysis with samples of East Asian populations (Fig. S25b), and the results largely recapitulated the geographical distributions of these populations, with Chinese Dai, Kinh Vietnamese, Southern Han Chinese, Northern Han Chinese, and Japanese individuals distributed alongside PC1.

We then counted the number of pSTRs per individual in the 26 populations. For each genome, the most abundant type of pSTR was dimeric (median of 19,024), followed by tetranucleotide (median of 7611) (Fig. 5a). We also found that Africans harbored the greatest numbers of pSTRs, corroborating the out-of-Africa model[78], while individuals of East Asia had the lowest number of pSTRs (Fig. 5a). In line with SNPs[79] and SVs[77], individuals of African ancestry exhibited a higher heterozygote/homozygote ratio of pSTRs than individuals from other ethnic groups (median of 3.43 versus 2.32) (Fig. 5b, c). The availability of genotypes of both SNPs and STRs enabled us to directly compare the diversity between SNPs and STRs in diverse populations. We computed the correlation between SNP heterozygosity and pSTR diversity, in which SNP heterozygosity[80] was the ratio of heterozygous bi-allelic SNPs over the length of the autosomes, and pSTR diversity was the average pairwise difference in pSTRs for each sample in each population. We observed that pSTR diversity had a strong positive correlation with SNP heterozygosity ($R^2$: 0.99, $P < 2.2E{-}16$) (Fig. S26), which was higher than that of mobile element insertions[43]. Africans showed the highest pSTR diversity and SNP heterozygosity, whereas both pSTR diversity and SNP heterozygosity of Americans showed great variability.

We next investigated the sharing of pSTRs across 26 different populations, and this analysis identified pSTRs shared in all populations ("All"), more than one ("Shared"), and specific to one population ("Unique") (Fig. 5d). For pSTRs detected in each population in the 1KGP dataset, over 40% of pSTRs could be found in all 26 populations, and the percentage of pSTRs unique to a specific population was less than 4%. Partly due to larger sample numbers (Southern Han Chinese in NyuWa: 600, Northern Han Chinese in NyuWa: 418), we detected more unique pSTRs in the two populations from the NyuWa dataset. Among these 24,426 NyuWa-specific pSTRs, 43 loci contained LoF alleles and 2998 loci were found in disease genes from Online Mendelian Inheritance in Man (OMIM) database, demonstrating the added value of the NyuWa dataset.

## pSTR length comparisons across populations

We then leveraged our call set to identify pSTRs with significant mean length differences between superpopulations, as these loci may contribute to the phenotypic differentiation between different populations. As samples of the 1KGP dataset were sequenced by one institution (the New York Genome Center) with a PCR-free protocol[44],

we restricted this analysis to the 1KGP dataset to avoid potential batch effects. As previously done with VNTRs[47], we performed pairwise length comparisons using the Wilcoxon rank-sum test among five continent populations for pSTRs with heterozygosity >0.1 and showed the results using volcano plots, with notable outliers labeled with the host genes of the loci (Fig. 6). These results well corroborated the population structure analysis above: the pSTR lengths of Africans showed substantial discrepancies with those of the other four superpopulations and Americans had STR length distributions similar to those of Europeans and South Asians. We also used the fixation index Rst[81] between different populations to quantify population differences in these pSTRs (Supplementary Data 9), and the results largely fit the findings of length comparisons, with the lowest average Rst between Americans and Europeans (mean Rst = 1.73%).

Based on the length comparisons, we selected four pSTRs frequently classified as top outliers among superpopulations to investigate further, and they resided in *UBE2L3*, *DHTKD1*, *MYPN*, and *MGAT5* (Fig. S27). The observed differences using the Wilcoxon rank-sum test were further validated by one-way ANOVAs and subsequent Tukey's multiple comparison test. The pSTR in the intron of *UBE2L3*, a member of the E2 ubiquitin-conjugating enzyme family, was mostly expanded in individuals from East Asia (Fig. S27a), and this pSTR was in strong LD with several GWAS SNPs implicated in multiple phenotypes, such as Crohn's disease, high-density lipoprotein cholesterol levels, and systemic lupus erythematosus (Supplementary Data 6). The pSTR in the intron of *DHTKD1* showed longer repeats in Africans than in any other, with an overall right-tailed length distribution (Fig. S27b). The *DHTKD1* gene encodes a dehydrogenase involved in mitochondrial energy production[82] and mutations in this gene have been associated with 2-aminoadipic 2-oxoadipic aciduria and Charcot-Marie-Tooth Disease Type 2Q[83,84]. Of note, we found that the pSTR in *DHTKD1* could regulate the expression of *DHTKD1* in LCL (beta = −0.287, adjusted $P = 3.61E{-}7$), suggesting that this pSTR may contribute to the expression differences of *DHTKD1* in different populations. The cumulative plot of the pSTR in *MYPN* revealed a bimodal distribution in East Asians and Africans, with longer repeats in East Asian populations (Fig. S27c). *MYPN* encodes striated muscle-specific sarcomeric protein, and mutations in *MYPN* have been associated with multiple cardiac diseases[85]. The pSTR in *MGAT5* gene, encoding an important enzyme involved in the synthesis of glycoprotein oligosaccharides, had longer repeats in individuals in Africa, and it showed a similar bimodal distribution in East Asians as the pSTR in *MYPN* (Fig. S27d).

We also compared the STR lengths among populations of East Asia, and the results largely resembled the population structure analysis, with great differences detected between Japanese individuals and other East Asians (Fig. S28). Previous SNP-based studies have observed genetic differences between northern (Han Chinese in Beijing, CHB) and southern Han Chinese (southern Han Chinese, CHS) individuals[42,86], but we failed to detect marked differences in pSTR length distributions between the two subgroups (Fig. S28), with no pSTRs below the significance level of 5% at all the loci tested. In line with this, the average Rst between northern and southern Han Chinese individuals was -0.29–0.34% across motif lengths (Supplementary Data 9).

## pSTR length variance analysis

In addition to the pSTRs with significant changes in mean lengths between superpopulations, we also considered the highly variable pSTRs in each superpopulation. Focusing on highly polymorphic pSTRs (pSTR with heterozygosity >0.1), loci with the top 5% variance within a population were considered highly variable. In total, we identified 4616 highly variable pSTRs, of which 1100 loci were detected in all superpopulations (Fig. 7a; Supplementary Data 10). GO enrichment analysis demonstrated that genes enclosing these loci played roles in the development and function of the nervous system (Fig. 7b).

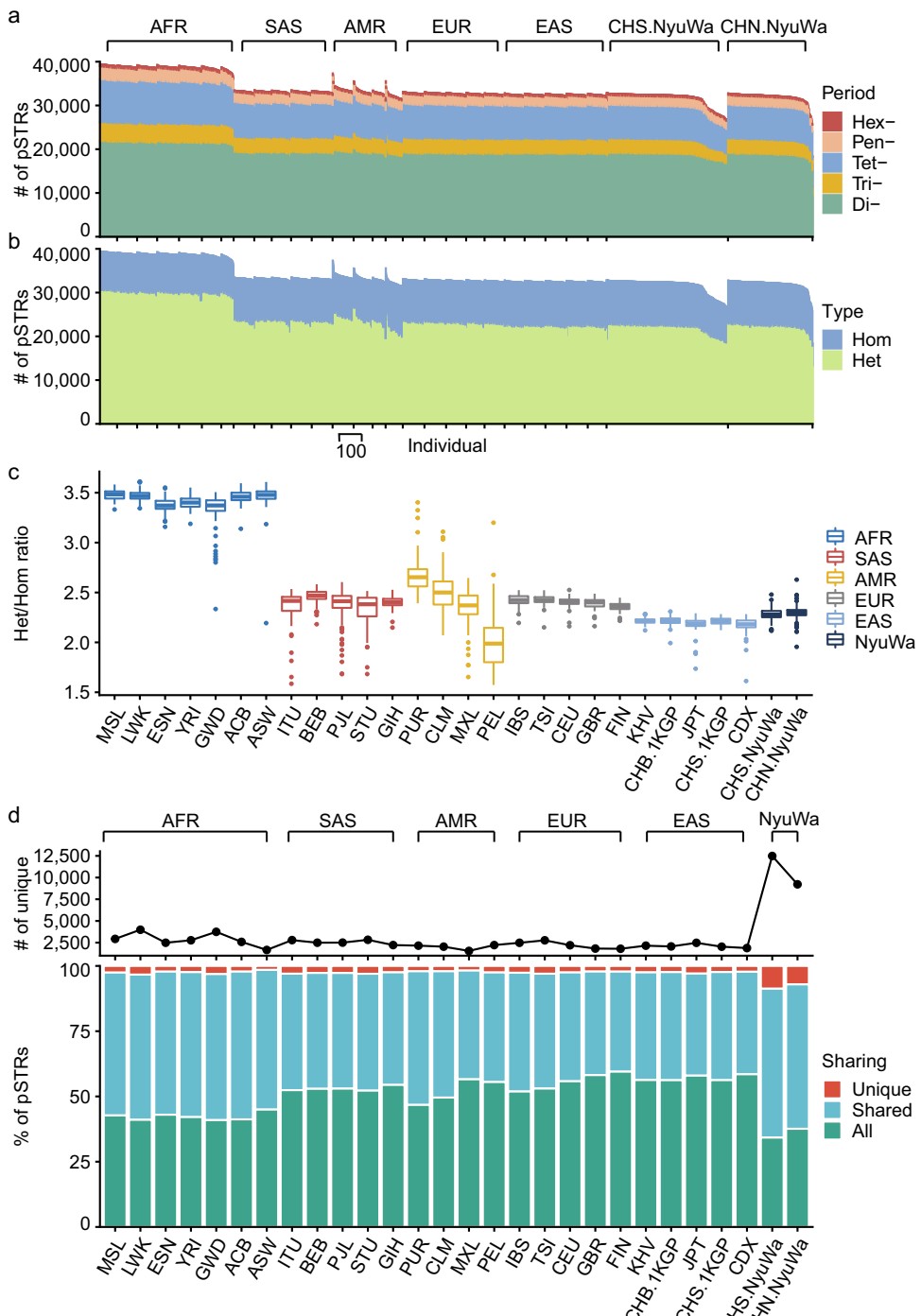

**Fig. 5 | pSTR counts per sample and population sharing across different populations. a**, **b** Number of pSTRs per individual stratified by motif length (**a**) or state (heterozygous or homozygous) (**b**) of pSTR loci. **c** Distributions of the heterozygote/homozygote ratio per individual (*n* = 3522) in populations of the 1KGP and NyuWa datasets. Horizontal lines indicate the median and boxes span from the lower quartile (the 25th percentiles) to the upper quartile (the 75th percentiles). Whiskers extend to points that are within 1.5 × IQR (interquartile range) from the upper or the lower quartiles. **d** Number of unique pSTRs (upper) and sharing of pSTRs (lower) across different populations of the 1KGP and NyuWa datasets. Unique, only exist in the corresponding population; Shared, exist in more than one but not all populations; All, exist in all populations. Abbreviations of populations are from 1KGP (Supplementary Data 1). CHN.NyuWa denotes Northern Han Chinese from the NyuWa dataset, and it is equivalent to Han Chinese in Beijing from the 1KGP dataset (CHB.1KGP). CHS.NyuWa denotes Southern Han Chinese from the NyuWa dataset, and it is equivalent to CHS.1KGP (Southern Han Chinese).

Concordantly, these genes were significantly enriched for genes predominantly expressed in the cerebral cortex (Fig. 7c). These results suggested that highly variable STRs may contribute to divergences in neurological phenotypes in humans. Among these highly variable loci, three pSTRs were tagged by GWAS SNPs implicated in nervous or immune system diseases, and their allele length distributions in different superpopulations were similar (Fig. 7d). STR "chr1:2609035" in the intron of *MMEL1* was tagged by multiple SNPs associated with sclerosis, autoimmune disease, and rheumatoid arthritis; STR "chr15:77705175" in the intron of *LINGO1* was in high LD with SNPs

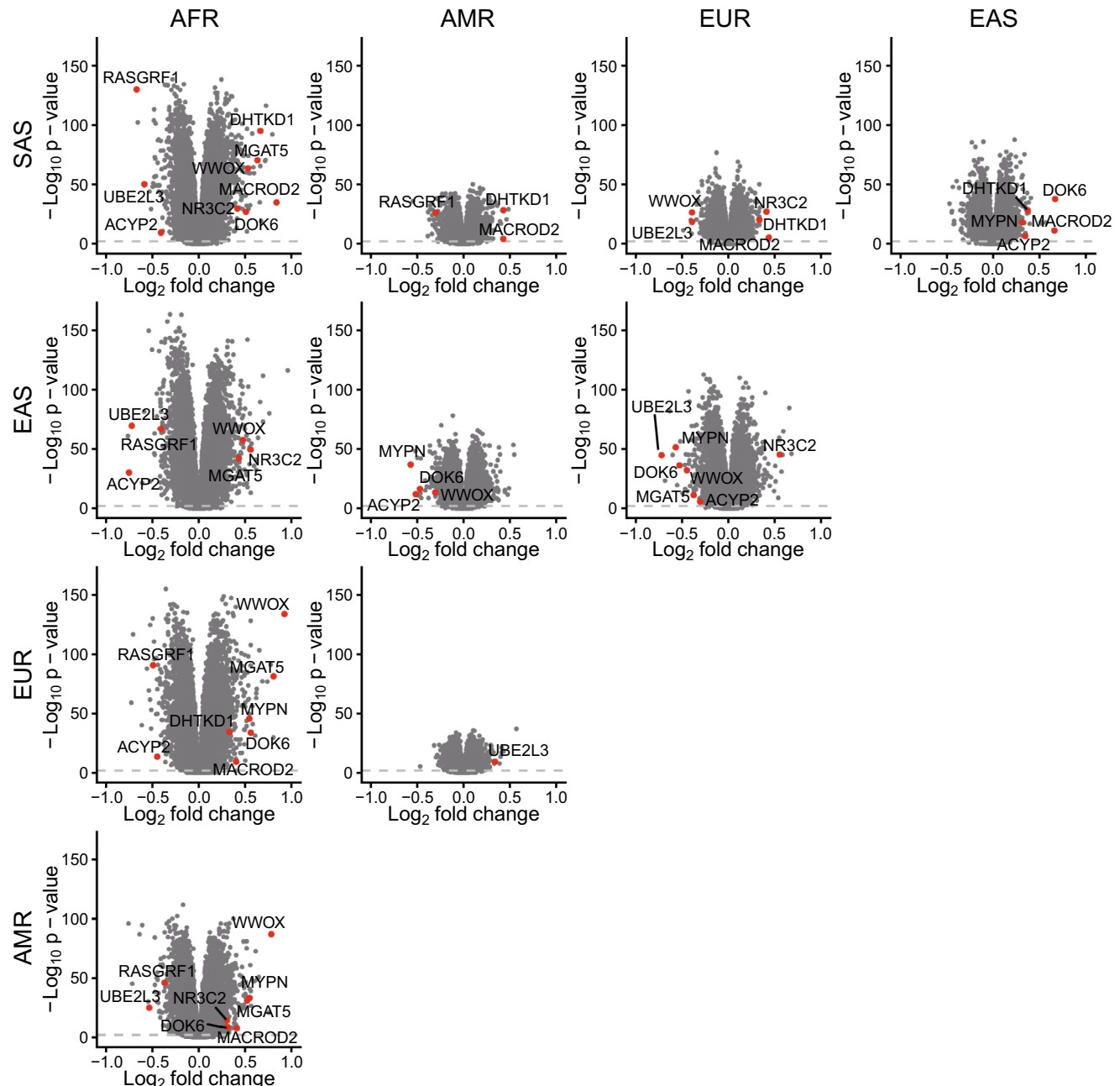

**Fig. 6 | Comparing STR lengths across the five superpopulations in the 1KGP dataset.** Pairwise comparisons of average pSTR lengths between superpopulations from the 1KGP are shown using volcano plots. pSTRs with top length differences were labeled using the genes in which they reside. The sample size of each superpopulation is 347–661. AFR African superpopulation, AMR American superpopulation, EAS East Asian superpopulation, EUR European superpopulation, SAS South Asian superpopulation. *P*-values were derived from two-sided Wilcoxon rank-sum test and adjusted using the Benjamini & Hochberg correction.

related to neuroticism measurement, unipolar depression, and mood disorder; STR "chr10:6586116" in the intron of *PRKCQ-AS1* was tagged by SNPs related to multiple traits such as asthma, eczema, and allergic rhinitis (Supplementary Data 6).

To further detect pSTRs with significant changes in variance of allele lengths among different superpopulations, we compared the standard deviations (SDs) of allele lengths of each site and used permutation test (1000 times) to get empirical *p*-values. Similar to pSTR mean length comparisons above, we performed this analysis in the superpopulations of the 1KGP dataset and pSTRs with heterozygosity >0.1. As shown in Fig. S29a, many loci showed differential variability in allele lengths between different superpopulations. Of note, we found that one known disease locus "chr12:50505002", at which GGC

expansion can cause FRA12A type of intellectual developmental disorder, showed lowest variance in East Asians (Fig. S29b). This STR was also in high LD with GWAS SNPs related to balding, body fat, and fibrinogen (Supplementary Data 6). Another locus "chr8:127102614", which was in intron of non-coding gene *PCAT1*, showed highest variance of allele lengths in African samples (Fig. S29b), and this site was in high LD with multiple GWAS SNPs associated with prostate carcinoma. STR "chr7:2812637" in intron of *GNA12* had lower length variance in East Asians and Africans than in other ethnical groups (Fig. S29b), and we found that this site could regulate the expression of *GNA12* in LCL (beta = −0.49, adjusted *P* = 4.36E−21). The *GNA12* gene encodes a subunit of the G proteins and functions as modulators or transducers in various transmembrane signaling systems[87,88]. In

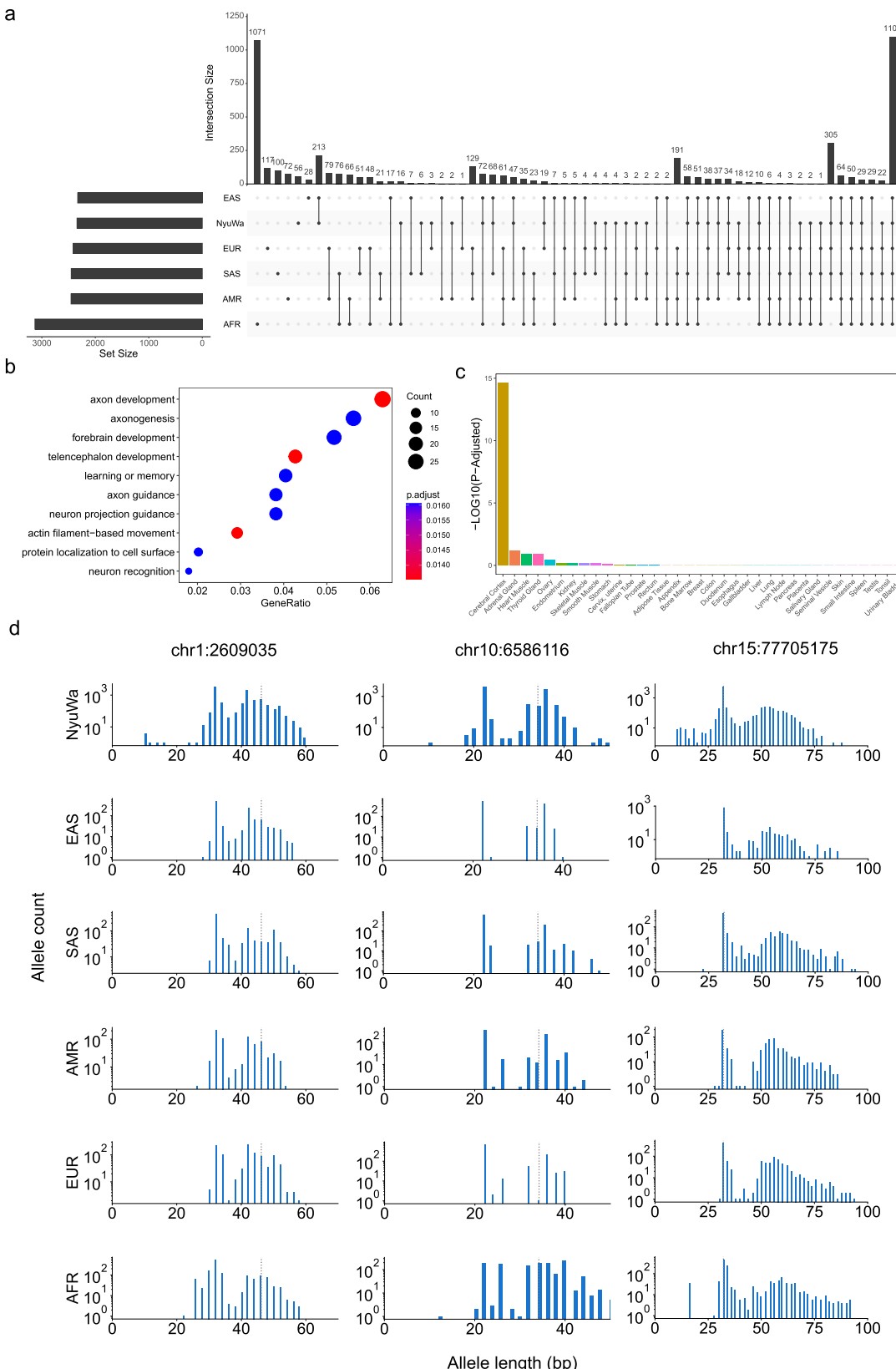

**Fig. 7 | Highly variable pSTRs within superpopulations. a** UpSet plot of highly variable pSTRs identified from the NyuWa dataset and the five superpopulations in the 1KGP. **b** Gene Ontology (GO) enrichment analysis for genes enclosing 1110 highly variable pSTRs detected in all superpopulations. The top ten most significant items are shown. **c** Bar plot showing the adjusted *p* value of tissue-specific gene enrichment. *P*-values were derived from hypergeometric test and corrected using the Benjamini & Hochberg correction by TissueEnrich v1.16.0. **d** Distribution of allele length for three examples of common highly variable pSTRs stratified by population. The gray vertical line indicates the reference allele length for the corresponding locus.

addition, intergenic STR "chr2:44076978", which had greatest length variance in African individuals (Fig. S29b), could regulate the 3′UTR alternative polyadenylation of multiple transcripts of *PREPL*, a member of the prolyl oligopeptidase subfamily of serine peptidases. These results together reveal another layer of population differentiation signature of pSTR.

## pSTR expansion analysis

Since dozens of STR expansions have been identified as causal variants in a range of human disorders[11], we next focused on detecting repeat expansions in our call set. We used a similar expansion score as described by Press et al.[57], which compared the 95th percentile and median of allele lengths of each pSTR. To mitigate the confounding of population differences, we performed this analysis in each superpopulation. Using this metric, we identified 213 pSTRs with an expansion score ≥2 in at least one superpopulation, mainly from di- and tetranucleotide STRs (Fig. S30a; Supplementary Data 11). Most expanded loci were found in intronic and intergenic regions, while none were in CDS. Notably, the pSTR "chr1:200669896" in the 5′UTR of *DDX59* was expanded in the NyuWa dataset and East Asians in the 1KGP and was in high LD with SNP rs6700559, which is associated with coronary artery disease (Fig. S30b). There were several other expanded STR loci tagged by GWAS SNPs, implying the potential influence of STR expansion on complex traits. For instance, the STR "chr2:193509528" in intergenic regions was tagged by GWAS SNPs associated with schizophrenia and a variety of neurological disorders, including Tourette syndrome and autism spectrum disorder (Fig. S30b). In addition, the STR "chr5:151073583" in the intron of *TNIP1* was in high LD with the GWAS SNP rs3792783, which is related to systemic scleroderma (Fig. S30b).

## STR variation in known disease loci

In humans, dozens of pathogenic STR loci have been found at which repeat number expansion above the locus-specific pathogenicity threshold could cause genetic diseases, which are called repeat expansion diseases and primarily affect the nervous system[89]. For the diagnosis of these diseases, whole genome sequencing (WGS) was increasingly implemented as a first-tier screening test and was a feasible approach because of advances in bioinformatics[90]. To characterize the variations in these STRs in our dataset, we analyzed 60 disease-causing STR loci that we collected (Fig. S1; Supplementary Data 2). After filtering the alleles by the number of supporting reads (method), we observed a repeat number distribution in the population similar to that reported by gnomAD v3.1.2[70] for each STR locus, including 7 STRs with at-risk expansion alleles (Fig. S31) and the others without such alleles. To verify the accuracy of genotypes, we compared them with the results from two previous studies (see "Methods") and observed high concordance at most loci (Fig. S32). Further manual inspection of the read visualizations confirmed 42 expansion alleles in total, which were equal to the threshold or longer than the threshold by one or two repeats. These results indicated a rare presence of at-risk expansion alleles in the natural human population. Allele distributions of these STRs were provided and could be used as a comparison set for studies of tandem repeat diseases.

## Discussion

Despite the plethora and high instability of STRs in the human genome, they have been more or less overlooked in most population genomic studies[3,4]. Accordingly, resources for pSTRs in diverse human populations are clearly lacking compared to other types of variants, such as SNPs and SVs. Moreover, to our knowledge, there are no population-level studies of genome-wide STR variation in Chinese individuals; thus, it is imperative to fill this gap. Here, using a state-of-the-art STR genotyper, we jointly analyzed the deep WGS 6487 genomes from the NyuWa and 1KGP datasets to perform a genome-wide interrogation of

more than 700,000 STRs. We described our STR call set and then analyzed the mutational patterns, functional properties, gene-regulatory effects, and expansion of pSTRs. The results of these analyses deepened our knowledge of STR biology. Then, we used pSTRs to implement population analysis and investigate population differentiated signatures. We also identified highly variable pSTRs within the superpopulation and explored their potential effects on complex human traits. We also genotyped 60 known disease-causing STRs to provide their allele distributions.

By combining the NyuWa dataset (3983 genomes) and the 1KGP dataset (2504 genomes), we constructed a large and high-quality resource of STR variation for diverse populations, especially for the Han Chinese population, which would fill the scarcity of STR variation resources in East Asian populations and benefit future STR studies. In total, our call set included 366,013 pSTRs and 290,454 mSTRs with 2–6 bp repeat motifs, in which 89,719 pSTRs were specifically identified in the NyuWa dataset. Within the pSTR call set, we found that 54.2% of pSTRs were highly multiallelic and that the STR mutation rates were influenced by motif length, chromatin context and genomic niche. We also observed a significant enrichment of hexameric pSTRs in subtelomeric regions and a modest correlation between hexameric pSTRs and double-strand breaks.

It is fundamental to know the functional impact of STR variation in humans. We found that trimeric and hexameric STRs were overrepresented in the 5′UTRs and CDSs. We found that pSTRs in CDSs tended to be less variable and were under strong selective constraint. We also identified 668 LoF STR alleles that could affect open-reading frames or transcript splicing. A total of 2871 pSTRs were found in high linkage disequilibrium with GWAS SNPs, which indicated a potential association between these pSTRs and complex human traits and diseases. Furthermore, we identified some pSTRs with gene-regulatory effects, including 3273 eSTRs and 1117 3′aSTRs. These pSTRs were enriched in regulatory elements and accessible chromatin regions. Later, we performed expansion analysis and identified 213 expanded pSTRs. These results extended our understanding of the functional impact of STR variation and provided a set of STRs with potentially functional impacts.

The large sample size and high population diversity of our dataset enabled us to conduct a population analysis based on STRs. The results of the population structure analysis suggested the power of pSTRs to distinguish populations. Using both STR and SNP genotypes, we observed a strong positive correlation between pSTR diversity and SNP heterozygosity. Later, we identified a set of population differentiated pSTRs based on length differences between superpopulations of the 1KGP dataset. We also found 1100 pSTRs that were highly variable in all superpopulations. Enrichment analysis suggested that these pSTRs potentially contribute to divergences in neurological phenotypes in humans.

Finally, we provided the allele distributions of 60 known disease-causing STRs and observed the rare presence of pathogenic expansions. Compared with previous studies, our allele distributions involved the genotypes of many Han Chinese individuals. Therefore, these distributions could serve as a comparison set for studies of tandem repeat diseases, especially for such studies in Chinese populations.

However, our study is limited by the STR genotyper we used. GangSTR v2.4.2 cannot analyze mono-nucleotide repeats resulting from their absence in the reference file and cannot analyze sex-chromosome STRs because the analysis of haploid genotypes was not supported. In addition, this tool was designed to analyze the repeat number variation of STR loci; hence, STR alleles with mutations inside the repeat units or interruptions remained undiscovered. Another limitation is that variations of nonreference (not included in the GangSTR reference file) STR loci have not yet been included. In the future, we would consider combining different STR identification and

genotyping tools, including reference-independent tools, to provide a broader description of STR variation in humans.

## Methods

### Samples and preprocessing

This study was approved by the Medical Research Ethics Committee of Institute of Biophysics, Chinese Academy of Sciences and complies with all relevant ethical regulations. All participants provided written informed consent. The informed consent is used to collect samples for genome studies conducted by Chinese Academy of Sciences. Deep WGS data in this study were collected from the NyuWa dataset (~31.5x)[42] and the 1KGP dataset (~33.3x)[44,52]. The NyuWa dataset contained 4013 unrelated individuals from different provinces in China[42], and was not ascertained for a specific health status. 2999 samples of the NyuWa dataset were reported before[42] and 1014 samples were newly sequenced using the Illumina platform. The GATK Best Practices Workflows germline short variant discovery pipeline[91] was employed to process and align reads to the human reference genome (GRCh38). For the 1KGP dataset, 2504 CRAM-format files mapped to the human genome build GRCh38 were downloaded from https://ftp.1000genomes.ebi.ac.uk/vol1/ftp/data_collections/1000G_2504_high_coverage/, which were recently sequenced at >30x coverage by the New York Genome Center[44]. The CRAM files were first converted to BAMs using SAMtools v1.14[92]. We then estimated the autosome coverage of all samples using mosdepth v0.3.3 (https://github.com/brentp/mosdepth). For the NyuWa dataset, the sexes of all samples were determined using guess-ploidy in BCFtools v1.10[93]. The sample information used in this study can be found in Supplementary Data 1.

### Genome-wide STR genotyping

Owing to its ability to genotype genome-wide STRs in a reliable and efficient way[49,50], we used GangSTR v2.4.2[48] to genotype autosomal STRs with option "--include-ggl". The reference file "hg38_ver13.bed.gz" for GangSTR were downloaded from https://github.com/gymreklab/GangSTR, and there were 765,227 autosomal STR loci (115,771 dinucleotide, 134,296 trinucleotide, 325,745 tetranucleotide, 155,589 pentanucleotide and 33,826 hexanucleotide STRs) retained after excluding sites with unit length >6 bp. The median length of these loci was 12 bp. 3994 samples from NyuWa dataset and 2504 samples from 1KGP dataset were successfully genotyped by GangSTR. The filtering steps for GangSTR calls were largely concordant with previous study[53]. DumpSTR in TRTools toolkit v4.1.0[94] was used for call-level filtering with options "--vcftype gangstr --zip --gangstr-min-call-DP 20 --gangstr-max-call-DP 1000 --gangstr-filter-spanbound-only --gangstr-filter-badC". Then we examined the call rate of each sample. 11 samples from NyuWa which had a call rate <50% were removed. Filtered VCFs of 6487 samples were merged using mergeSTR in TRTools toolkit v4.1.0[94] with default parameters, and merged VCF was then subjected to locus-level filtering by dumpSTR with parameters "--filter-regions GRCh38GenomicSuperDup.bed.gz --filter-regions-names SEGDUP --min-locus-callrate 0.2 --min-locus-hwep 0.00001". The file "GRCh38GenomicSuperDup.bed.gz" were downloaded from the UCSC Genome Browser database[95] and was to remove sites overlapping segmental duplications. Then the remaining sites with more than one allele were considered as polymorphic STRs (pSTRs), otherwise as monomorphic STRs (mSTRs). The final call set consisted of 366,013 pSTRs and 290,454 mSTRs from 6487 genomes. Some per-locus statistics including maximum observed allele length, heterozygosity, bit-entropy of the distribution of alleles, mean allele length, mode allele length, and variance of allele length were computed by statSTR in TRTools toolkit v4.1.0[16].

### Genotyping known pathogenic loci

There were no known cases of STR disease for samples in this study. To supplement the genome-wide analysis of GangSTR, we collected 60

known disease STR loci (Supplementary Data 2) from multiple sources including ExpansionHunter[51], STRipy[96], Stranger (https://github.com/Clinical-Genomics/stranger), and gnomAD (https://github.com/broadinstitute/str-analysis). ExpansionHunter was designed to targeted genotyping analysis. It had good correlations with PCR-based assays[90,96] and has been successfully used in several large-scale cohorts[90,97,98]. We used ExpansionHunter v5.0.0[51] to genotype 51 "standard" or "imperfect GCN" loci across all 6487 samples with default options. For nine "replaced/nested" loci (RFC1, BEAN1, DAB1, MARCHF6, RAPGEF2, SAMD12, STARD7, TNRC6A, and YEATS2)[96], we used "call_non_ref_pathogenic_motifs" in str-analysis v0.9.4 (https://github.com/broadinstitute/str-analysis) to genotype all samples, as ExpansionHunter was unable to differentiate between the pathogenic and reference repeat motifs[96]. We then merged the calls from ExpansionHunter and "call_non_ref_pathogenic_motifs", and filtered out alleles with fewer than five total supporting reads. For all individuals at each locus, we used REViewer v0.2.7[99] to visualize alignments of reads if the inferred repeat length exceeded the normal range or pathogenic threshold of this locus. We then manually inspected each image for read visualization to validate the authenticity of these events.

There were seven disease loci which were also in GangSTR call set (Supplementary Data 3). To compare the calls between GangSTR and ExpansionHunter/str-analysis, we defined genotype concordance metric as Saini et al.[100]. The genotype concordance $c_i$ was: 1 if both alleles match; 0.5 if only one allele matches; 0 if neither alleles match. Then genotype concordance for a STR locus is the average over all the samples: $C = \frac{1}{n}\sum_{i=1}^{n} c_i$.

To compare the allele distributions of our call set with those from STRipy database[96], and gnomAD database v3.1.2[70] (https://gnomad.broadinstitute.org/short-tandem-repeats?dataset=gnomad_r3), we downloaded the allele distributions from https://gitlab.com/andreassh/stripy-pipeline and https://gnomad.broadinstitute.org/downloads#v3-short-tandem-repeats, respectively. We then plotted the distributions of each loci using box plots (Fig. S32).

### Mendelian inheritance rate

To validate the pipeline of genome-wide STR analysis by GangSTR, we used 60 trio families (in-house WGS data from Han Chinese) to evaluate Mendelian inheritance. We first generated STR calls for these samples using the pipeline above. For each trio, a STR locus was considered to follow Mendelian inheritance if one allele of the child's genotype was same to any one allele of the father's genotype and another one was same to any one allele of the mother's genotype, and the proportion of genome-wide STR loci that followed Mendelian inheritance was defined as Mendelian inheritance rate.

### Subtelomeric enrichment of STRs

We first cut each chromosome arm (p arm and q arm) into 1 Mbp consecutive windows, and removed windows overlapped with centromeres. Then the number of pSTRs/mSTRs in each window was summed. The mean STR number in the subtelomeric bins on each chromosome arm (outermost 5 Mbp of each chromosome arm end) was compared to the mean number of STRs in all other bins to get the fold increase, as described in ref. 62. Acrocentric arms (13p, 14p, 15p, 21p, and 22p) were not considered. We then performed a 10,000 round permutation test by exchanging the bin sums, and counted the number of events with permuted fold increase greater or equal to the observed fold increase. We divided this number by 10,000 to get the empirical P-value.

### Chromosome-level analyses of STR density

To check the distributions of pSTRs and mSTRs across the genome, we adopted the method by Collins et al. which has been used to investigate the chromosome density of structural variants[60]. For 22 autosomes, we first segmented each chromosome into 100 kb consecutive

bins and removed bins overlapped with centromeres. Then we counted the number of STRs (pSTRs or mSTRs) in each bin and smoothed the count using an 11-bin (~1 Mbp) rolling mean for each chromosome. For comparison of chromosome context, each bin was assigned to a percentile based on the position of that bin on its respective chromosome arm relative to the centromere (0: centromere; −1: p-arm telomere; 1: q-arm telomere). Next, the normalized bin positions (i.e., −1 to 1) were cut into 500 uniform intervals, and values across all autosomes based on the normalized interval position were averaged to generate "meta-chromosome" density. Finally, the "meta-chromosome" density was normalized by its mean value to get the "fold-enrichment" values, as shown in Figs. S9, S10, and S11. To compare the density of STRs between different chromosome contexts, we considered normalized positions within the outermost 5% of each chromosome arm as "telomeric", the innermost 5% as "centromeric" and the other 90% of each arm as "interstitial". The "fold-enrichment" scores in the given chromosome context were subjected to test if the values were greater or smaller than 1 with Student's t-test. We adjusted P-values using the Bonferroni method and the significant level was set to 5%.

### Correlation analysis of STR and genomic features

To correlate the pSTR occurrences with genome features, we used the methods described by Kojima et al.[101], which has been used to study mobile element variants. We repeated the procedure here for clarity. We first segmented the 22 autosomes into 1 Mbp consecutive bins and removed bins overlapped with centromeres. We counted the number of pSTRs/mSTRs in each bin. Next, we examined the correlations of pSTRs/mSTRs count with various genome features, such as GC percent, CpG count, gene count, A/B compartment, DNase hypersensitive sites, histone modifications, transcription factor (TF) binding sites, DNA methylation level, and replication timing. The public data used here can be found in Supplementary Data 4.

**GC percent.** "nuc" function in BEDTools v2.27.1[102] was used to get GC percent of each 1 Mbp genome bin.

**CpG count.** CpG island regions of the human GRCh38 genome were downloaded from UCSC Genome Browser database (accessed at 2022/04/16)[95] and the count of CpG in each genome window was generated using bedmap in BEDOPS v2.4.40[103].

**Gene count.** Using GNECODE v39[104], we counted the number of all genes and protein-coding genes across 1 Mbp windows. For all genes, genes annotated as "level 3", "TEC" or "pseudogene" were excluded.

**A/B compartment.** Two mcool format files of H1-hESC Hi-C data[105] were downloaded from NCBI GEO[106]: GSM5057489, formaldehyde fixation followed by HindIII; GSM5057481, formaldehyde and DSG fixation followed by HindIII cleavage. Then, we used GENOVA v1.0[107] to get compartment scores under 1 Mbp resolution.

**DNase hypersensitive sites.** DNase-seq peak files of H1-hESC cell and H9-hESC cell were downloaded from the ENCODE project[108] (Accession number: ENCFF905XDS, ENCFF574LKL, ENCFF338KTY, and ENCFF190JAO). We computed the average number of peaks in each 1 Mbp window of these three datasets.

**Histone modifications and TF-binding sites.** We counted the number of peaks of histone protein modifications in H1-hESC and H9-hESC and several TFs (CTCF, phospho-Pol-II A, Pol-II, and EP300) in H1-hESC from the ENCODE project (accession number in Supplementary Data 4)[108]. The number of peaks in each 1 Mbp bin was counted and the average number was used for analysis if replicates were available.

**DNA methylation level.** DNA methylation of three types of sites was considered in H1-hESC: CpG, CHG, and CHH. DNA methylation of two types of sites was considered in H9-hESC: CpG and CHG. Methylation state datasets in bigBed format of whole genome bisulfite sequencing (WGBS) were retrieved from the ENCODE project (accession number in Supplementary Data 4)[108]. We used methylKit v1.16.1[109] to read and compute the average methylation level across all sites in each 100 kb genome bin. Only sites with at least five reads were included for analysis. To control the outliers, we excluded genome bins with fewer than 500, 2500, and 10,000 CpG, CHG, and CHH sites, respectively.

**Replication timing.** Replication timing (RT) was calculated using the Repli-Chip datasets of H1-hESC cell from the ENCODE project (Accession number: ENCFF000KUF, ENCFF000KUG, and ENCFF000KUH)[108]. As the original datasets were mapped to human GRCh37 reference genome, we first performed liftover of probe positions from GRCh37 to GRCh38 for each dataset. The RT level across probes in each bin and across replicates were then averaged.

### Functional annotation

We annotated pSTRs using Variant Effect Predictor v99.2 (VEP)[110] with Ensembl database version 104[111], with parameters "--pick --sift b --polyphen b --hgvs --symbol --canonical --biotype --protein --domains --uniprot --tsl --numbers --distance 2000,1000". Gene set enrichment analysis for genes with trimeric and hexameric pSTR in their CDS and 5' UTR regions was performed by clusterProfiler v3.18.0[112]. Alleles annotated in the following categories were considered loss-of-function ones: frameshift mutations, splicing region variants, coding sequence variants, and start or stop codon variants. The enrichment of pSTRs across different genomic features was quantified by GAT v1.3.4[113] with 1000 permutations.

We also annotated pSTRs by their linkage disequilibrium (LD) with GWAS risk SNPs. GWAS risk SNPs with P-value $\leq 5 \times 10^{-8}$ and their related traits were obtained from GWAS Catalog v1.0.2[24]. GWAS SNPs that related to only "educational attainment", "mathematical ability" and "intelligence" were removed because these traits were strongly associated with environmental and socioeconomic factors. For the remaining GWAS risk SNPs, we first converted the genotype data into indicator matrix using PLINK v1.9[114]. LD between STR-SNP pair was defined as the squared Pearson correlation between STR dosage and SNP dosages, as described before[100]. STR dosage was defined as the sum of the repeat number of two alleles. Only pSTRs within 250 kb of a GWAS risk SNP were considered in LD calculation, and STR genotypes seen less than three times were removed, as described previously[16]. STR-SNP pair with $r^2 \geq 0.7$ was considered as strong LD. A complete set of pSTRs tagged by GWAS risk SNPs can be found in Supplementary Data 6.

### Identification of eSTR

To detect expression differences among individuals with different STR genotypes, BAMs files of 462 lymphoblastoid cell lines (LCLs) from Geuvadis consortium[71] were downloaded from: https://ftp.1000genomes.ebi.ac.uk/vol1/ftp/data_collections/geuvadis/working/geuvadis_topmed/. Of these samples, 445 samples overlapped with the 2504 1KGP genomes set (CEU: 89, FIN: 92, GBR: 86, TSI: 91, YRI: 87). These BAM files were produced by the TOPMed RNA-seq pipeline (https://github.com/broadinstitute/gtex-pipeline), with reads aligned to the human GRCh38 reference genome using STAR v2.5.3a[115]. We applied featureCounts v2.0.3[116] to count reads of genes with GENCODE v34 annotation[104]. Next, we generated log2 normalized FPKM values using edgeR v3.32.1[117], after filtering lowly expressed genes by "filter-ByExpr" function in edgeR. To account for hidden batch effects and other unobserved confounders, we used covariates for sex and population structure and detected additional hidden factors using Iteratively Adjusted Surrogate Variable Analysis (IA-SVA)[118], as

described before[46]. For population structure, we used the top ten principal components determined from the principal component analysis (PCA) of the SNP genotypes from the 445 individuals. Only SNPs with minor allele frequency ≥ 0.05 were used for PCA; pruning of 27 known long-range LD regions[119] and subsequent PCA were performed by PLINK v1.9[114]. For IA-SVA, we used sex and top ten genotype principal components as the known covariates to estimate a set of latent covariates for the expression values. We chose fifteen hidden covariates based on the correlations between covariates (Fig. S18). Finally, expression values were adjusted for sex, population structure and hidden factors using a linear model. Adjusted expression matrix was used for eSTR identification. This left 15,627 genes expressed in the LCL dataset.

The eSTR identification method was similar to previous studies[15,16]. Only STRs within 500 kb of a gene expressed in the LCL dataset were included. To ensure the site quality and the eSTR detection power, we kept loci where at least 50 of the 445 samples had a genotype call and loci with heterozygosity ≥0.1. We also filtered genotypes of each pSTR locus to control for outliers by removing any genotypes seen fewer than three times out of 445 samples. If there were less than three genotypes after filtering samples, the STR locus was discarded[16]. Finally, we converted the STR genotype calls to dosage (sum of the deviations in the STR allele repeat numbers from the reference allele repeat number). The final STR dosage matrix contained 39,933 sites.

For each gene-STR pair, a linear model as expression ~ dosage was used to test the associations between STR dosage and gene expression. As described before[16], we applied a gene-level FDR threshold of 10% to determine significant STR-gene pairs. For each gene, we first adjusted the P-values from association test with multiple STRs using Bonferroni method and we selected the lowest P-value. Then we used the P-values (one per gene) from all genes as input to "p.adjust" function in R to get final Q-values for each gene, with "BH" method. At last, we used alpha level of 10% to get all significant STR-gene pairs. We also repeated the above procedure after shuffling the sample identifiers, as a negative control.

### Identification of 3'aSTR

Starting with the 445 LCL BAM files mentioned above, we first converted them to bedGraph files using BEDTools v2.27.1[102], and only uniquely mapped reads were included. We then used DaPars v2.0 algorithm[120] to get a percentage of distal poly(A) site usage index (PDUI) value for each transcript in each sample under default settings. Similar to expression data processing, we included sex, population structure, and hidden IA-SVA factors to adjust the PDUI values. To remove lowly expressed genes, we only considered genes retained in eSTR analysis. The adjusted PDUI data was then used for 3'aSTR identification, containing 44,742 transcripts.

Using the normalized STR dosage data above, we adopted a similar method to test the associations between STR genotypes and transcript PDUI values. For each transcript-STR pair, we used linear regression to test their association and controlled FDR of 10% at transcript-level. Similarly, we also permuted the sample identifiers to repeat the association test for each transcript-STR as a negative control.

### Genomic context enrichment analysis of eSTR and 3'aSTR

To examine the enrichment with epigenetic features of eSTR and 3'aSTR, we downloaded the peak files of ChIP-seq of histone marks and CTCF and ATAC-seq of the GM12878 cell line from the ENCODE project[108] (Accession number: ATAC-seq, ENCFF748UZH; CTCF, ENCFF796WRU; H3K4me3, ENCFF998CEU; H3K27me3, ENCFF291DHI; H3K9ac, ENCFF981JOU; H3K27ac, ENCFF023LTU; H3K4me1, ENCFF321BVG; H3K36me3, ENCFF432EMI; H3K4me2, ENCFF283LNH; H3K9me3, ENCFF725UFY). To validate the results of GM12878 cell line, we also

downloaded peak files of DNase-seq, CTCF ChIP-seq, histone ChIP-seq of GM06990 cell line (Accession number: DNase-seq, ENCFF239QAE; CTCF, ENCFF031WEA; H3K27me3, ENCFF307CEJ; H3K4me3, ENCFF253XQQ; H3K36me3, ENCFF884ZOW) and GM12865 cell line (Accession number: DNase-seq, ENCFF754VPH; CTCF, ENCFF541DDH; H3K4me3, ENCFF438JMP). In addition, we also downloaded chromatin states data of these three cell lines from EpiMap[121], which were in 18 categories defined by ChromHMM[72].

To quantify the enrichment of eSTRs and 3'aSTRs in these epigenetic and genomic features, we adopted GAT v1.3.4 to get the fold enrichment value and empirical P-values. Then the P-values in each analysis were adjusted by "p.adjust" function with "BH" method in R and a significant level was set to 5%.

### GWAS hit enrichment analysis of eSTR and 3'aSTR

By randomly sampling n STRs from the STRs in eSTR/3'aSTR analysis 1000 times, where n was equal to the number of observed eSTRs or 3'aSTRs, we counted the times in which number of STRs tagged by GWAS lead variants was larger than actual number of eSTRs/3'aSTRs in LD with GWAS lead variants, and computed an empirical P-value (Fig. S24).

We also tested that whether eSTR/3'aSTR-associated genes were enriched in genes implicated by previous GWAS listed in GWAS catalog[24]. We first retained GWAS signals with P-value <$5 \times 10^{-8}$. For each trait, we retrieved the relevant genes from the columns "Reported Gene(s)" and "Mapped_Gene" and trait with fewer than 10 genes that were excluded, as described before[15]. We then performed the gene set enrichment analysis using clusterProfiler v3.18.0[112] and showed the top traits in Fig. S24.

### SNP heterozygosity and STR diversity

SNP heterozygosity was defined as the ratio of heterozygous SNPs over the length of the human genome[80], and the mean value was computed when multiple samples were included, as described in previous studies[43,122]. STR diversity was computed as the average number of STR differences between every two individuals in a given population. For the NyuWa dataset, high-quality bi-allelic SNP calls were used for SNP heterozygosity calculation. For deeply sequenced 1KGP dataset, SNP calls were downloaded from https://ftp.1000genomes.ebi.ac.uk/vol1/ftp/data_collections/1000G_2504_high_coverage/working/20201028_3202_phased/ and only bi-allelic SNPs were included. The number of heterozygous autosomal SNP numbers for each sample was computed by VCFtools v0.1.16[123]. STR diversity was computed by "gtcheck" function in BCFtools v1.7[93].

### Principal component analysis

To perform PCA, STR genotypes for each sample were first converted into dosage data (sum of repeat number of two alleles). Only pSTRs with major allele frequency <0.95 were used for each PCA, and PCA was done by using "prcomp" function in R. To match the sample number of NyuWa populations to the sample number of 1KGP populations, we sampled 100 individuals from Northern Han Chinese (CHN) and 100 samples from Southern Han Chinese (CHS) of the NyuWa dataset. We repeated three times of the sampling procedure and found that the PCA results (Fig. S25) were stable.

### Comparison of pSTR lengths across superpopulations

To identify pSTRs that were significantly different in mean length across superpopulations, we first combined the repeat numbers of two alleles for each STR locus in each sample. This analysis was restricted to the 1KGP dataset to avoid potential batch effect. To restrict to highly polymorphic STRs, we filtered out loci with heterozygosity <0.1. For each site, we then used Wilcoxon rank-sum test to compare the length distributions between each two superpopulations. We adjusted the P-values using the "p.adjust" function in R with "BH" method.

Comparisons with adjusted $P < 0.01$ were considered to be significant. The results were shown using volcano plots (Fig. 6).

### Rst levels

We used the definition of Rst from a previous study[81]. The fixation index Rst between populations for pSTRs was defined as: $Rst = \frac{S_t - S_w}{S_t}$, where $S_t$ was the average variances of allele length in all individuals under investigation, and $S_w$ was the average variances in allele length within each population. Only pSTR loci with heterozygosity >0.1 were used for Rst calculation. For allele length of each locus, the repeat number of two alleles was combined.

### Identification of highly variable pSTRs within superpopulation

For each pSTR in each superpopulation, we computed the variance of the loci among all individuals, and loci with top 5% variances were considered highly variable. In this analysis, only pSTRs with heterozygosity >0.1 (in each superpopulation) were included, as these loci had higher levels of polymorphisms. Gene set enrichment analysis for genes enclosing these pSTRs was performed by clusterProfiler v3.18.0[112]. Enrichment of these genes for tissue-specific genes was conducted in TissueEnrich v1.16.0[124].

### Comparison of pSTR variances across superpopulations

To identify pSTRs with significant change in variance of allele lengths, for each site, we calculated the differences of standard deviations (SD diff) of allele length in any two superpopulations of the 1KGP dataset. We restricted this analysis to pSTRs with heterozygosity >0.1. To compute empirical p-values, we randomly shuffling the sample labels and then calculated the fraction of times the absolute value of original SD diff is less or equal to the SD diff generated by the shuffled samples. In this study, we perform 1000 permutation per site. We adjusted the P-values using the "p.adjust" function in R with "BH" method. Comparisons with adjusted $P < 0.01$ were considered to be significant. The results were shown using volcano plots (Fig. S29).

### Expansion analysis of pSTR loci

In each superpopulation of the 1KGP dataset, we employed a similar expansion score metric as Press et al. previously described[57] to detect expanded loci. For each STR locus, the expansion score was defined as (95th percentiles of allele length − median allele length)/median allele length. Then we define expanded loci as those with expansion scores ≥2[57] in each superpopulation.

### Statistical analysis

All of the statistical analyses in this study were briefly described in the main text and performed using R v4.0.3[125].

### Reporting summary

Further information on research design is available in the Nature Portfolio Reporting Summary linked to this article.

## Data availability

The DNA sequencing data of NyuWa samples used in this study have been deposited in the Genome Sequence Archive (GSA) in National Genomics Data Center, China National Center for Bioinformation/Beijing Institute of Genomics, Chinese Academy of Sciences, under accession number HRA004185 (https://ngdc.cncb.ac.cn/gsa-human/). These data are available under restricted access for privacy protection and can be obtained by application on the GSA database website (https://ngdc.cncb.ac.cn/gsa-human/) following the guidance of "Request Data" on this website. These data have also been deposited in the National Omics Data Encyclopedia (NODE) of the Bio-Med Big Data Center, Shanghai Institute of Nutrition and Health, Chinese Academy of Sciences, under accession number OEP002803 (http://www.

biosino.org/node). The user can register and login to this website and follow the guidance of "Request for Restricted Data" to request the data. A full list of pSTRs generated in this study has been deposited in the Genome Variation Map (GVM) in National Genomics Data Center, China National Center for Bioinformation/Beijing Institute of Genomics, Chinese Academy of Sciences, under accession number GVM000464. The user can contact the corresponding author to apply for permission to access this data. The reference genome GRCh38 used in this study is available at https://console.cloud.google.com/storage/browser/genomics-public-data/resources/broad/hg38/v0/. The alignment files for the 1KGP dataset are available at https://ftp.1000genomes.ebi.ac.uk/vol1/ftp/data_collections/1000G_2504_high_coverage/. Genotype data for SNPs and indels for the 1KGP dataset is available at https://ftp.1000genomes.ebi.ac.uk/vol1/ftp/data_collections/1000G_2504_high_coverage/working/20201028_3202_phased/. RNA-seq data of the GEUVADIS Project is available at https://www.internationalgenome.org/data-portal/data-collection/geuvadis. The chromatin states data for GM12878, GM06990, and GM12865 cell lines is available at https://personal.broadinstitute.org/cboix/epimap/ChromHMM/observed_aux_18_hg38/CALLS/. GWAS Catalog variants are available at https://www.ebi.ac.uk/gwas/docs/file-downloads. Results of STRs in LD with GWAS SNPs, QTL analyses, and expansion analysis generated in this study are provided in the Supplementary Data file and are also available from a public website (http://bigdata.ibp.ac.cn/STR).

## Code availability

Analysis scripts for reproducing the analysis and figures in this study are provided in the GitHub repository: https://github.com/YiweiNiu/STR_2022/releases/tag/v0.1.

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

## Acknowledgements

We thank the people for generously contributing samples and sequencing data to the NyuWa dataset and the 1KGP dataset. Data analysis and computing resources were supported by the Center for Big Data Research in Health (http://bigdata.ibp.ac.cn), Institute of Biophysics, Chinese Academy of Sciences. This work was supported by Strategic Priority Research Program of the Chinese Academy of Sciences [XDB38040300 (S.-M.H.)]; National Key R&D Program of China [2021YFF0703701 (S.-M.H.), 2021YFF0704500 (P.Z.)]; 14th Five-year Informatization Plan of Chinese Academy of Sciences [CAS-WX2021SF-0203 (S.-M.H.)]; National Natural Science Foundation of China [91940306 (S.-M.H.), 31871294 (S.-M.H.), 31970647 (P.Z.), 32200478 (Y.-Y.L.)]; Special investigation on science and technology basic resources of the MOST, China [2019FY100102 (P.Z.)]; China Postdoctoral Science Foundation [2022M713311 (Y.-Y.L.)]; and National Genomics Data Center, China.

## Author contributions

T.X. and S.M.H. conceptualized and supervised the project. Y.R.S., Y.W.N., P.Z, H.X.L., S.L., S.J.Z, J.J.W., Y.Y.L., X.Y.L., and T.R.S. conducted data analysis. Y.R.S., Y.W.N., P.Z., H.X.L., T.X., and S.M.H. drafted the manuscript, and all the primary authors reviewed, edited, and approved the manuscript.

## Competing interests

The authors declare no competing interests.
