## [Peer Review File · Nature Communications]

Characterization of genome-wide STR variation in 6,487 human genomesREVIEWER COMMENTS

Reviewer #1 (Remarks to the Author):

Summary:

The authors present a comprehensive analysis of short tandem repeats (STRs) in a set of 6,487 human genomes which all originate from Asia. STRs are the second most common type of polymorphisms in the human genome, after SNPs and indels. Despite their abundance these variants have not been used extensively in genetic analysis after the advent of short read sequencing. Their underutilization despite their high and valuable information content is mainly due to the algorithmic complexities of analyzing these variants from short read data.

The current manuscript adds to a limited set of publications exploring the patterns of polymorphism of this important variant class and roughly 370 thousand polymorphic STRs (pSTRs) are identified out of the 765 thousand STRs that are analyzed, the rest are monomorphic (mSTRs).

The main novelty of the paper is the origin of the sample set since it is the largest analysis to date containing only samples from Asia.

Previously reported patterns for mutational behavior and chromosomal distribution of STRs are replicated both with respect to chromosomal position and functional annotation. Expansions are determined at 113 pSTRs using a previous definition of expansion status, including one pSTR in a 5'UTR region and in LD with a SNP associated with CAD which merits further research. A population analysis of the pSTRs recapitulates previous patterns and separates both super and subpopulations in an expected manner.

An interesting part of the analysis is an allele length distribution comparison between the populations and the analysis of four frequent outlier pSTRs which I feel deserves further analysis and more discussion.

The last two parts of the analysis focus on highly variable pSTRs and known pathogenic pSTRs. The genes enclosing the highly variable pSTRs are shown to be enriched for genes expressed in the cerebral cortex and genes involved in the development and function of the nervous system. For the known pathogenic pSTRs, a small subset of samples alleles equal to or longer than the pathogenicity threshold were observed but no comments are made as to whether the carriers of the alleles suffer from the conditions associated with the expansions.

The following parts of the paper display creative analysis and could be extended and emphasized:

*Expansion analysis (using definition from Press et al.)

*Between population length distribution comparison and analysis of frequent outliers

*Highly variable pSTRs with length distributions that are different between populations

The parts that mostly confirm previously reported results should possibly be moved to a supplement to better enable readers to detect the novel results presented.

The sample set is large but no phenotypes are used for associating the genotypes generated, whether that is due to their unavailability is not addressed and I feel association results would add weight to the paper. Further, despite the fact that the set is larger than previously published set of Asian origin, large parts of the analyses are performed on subsets of the full set or only on samples with origins outside of Asia.

Major Comments:

1. Would suggest proof reading of the English in the paper, in some cases this does not affect understanding but in others, sentences are hard to parse.
2. It is true that most individuals in the UKB study were white British but the South Asian population was 3063 individuals (only 1000 samples less than the NyuWa set) and the African population was 2976 samples.
3. Why are there only seven sites (out of 60 known pathogenic) that both GangSTR and Expansion hunter genotype?
4. This analysis should be repeated per motif length and per major allele length to confirm bias of

longer alleles towards contraction and shorter to expansion (line 170)

We observed a higher proportion of alleles longer than the major allele (45.2%) compared to alleles shorter than the major one (31.3%), and the allele frequency monotonically decreased with deviation from the major allele (Fig. 2f; Fig. S6c), coinciding with the stepwise mutation model [55].

5. 0.83% of the polymorphic STRs are shown to have an $R^2 \geq 0.7$ with a SNPs associated with traits or diseases but no further analysis to determine causality was performed. Were these pSTRs located in more functionally important regions than the SNPs they tag f.ex? Or is there some way to do conditional association with the phenotypes that the tagged SNPs are associated with?

6. A set of 445 samples was used to demonstrate associations of dosages of specific STRs with expression and 3'UTR alternative polyadenylation of genes close by. These samples were however of European and African ancestry, different from the focus of the paper. This analysis seems out of place compared to the other parts of the paper which focuses mostly on characterizing pSTRs in Asian populations. Further, similar analysis has already been published for 311 out of the 445 samples used here, reporting similar results and is cited in the paper.

7. How is the frequently used 10% FDR chosen? -> This choice should be rationalized and justified either in Methods or Supplement

8. I suggest making more use of the trio families, a locus that does not follow Mendelian inheritance could harbor a true microsatellite mutation. Could order violating loci by qq scores and examine them in decreasing order to deny or confirm the existence of a mutation

9. Defining STR dosage as the sum of the repeat number of the two alleles is valid but since $10+10=18+2$, I would suggest adding the same analysis with STR dosage defined as the length of the allele further from the ref length or longer allele. Some proteins could have an eSTR with a threshold for which all alleles above or below block expression, this merits thresholding alleles before doing regression. Doing single allele R^2 would also work, i.e, creating binary marker for each allele.

10. Why use the dosage effect (sum of two alleles) in length comparison between populations? I suggest weighing each allele length with its frequency in the population to obtain a more accurate value for average STR length per population.

Minor comments

1. It needs to be clearer that both Beyter et al. and Mukamel et al. cited in the introduction cover only VNTRs (motifs ≥ 7 bp), not STRs.

2. Line 126 should reference Fig. S2e but not Fig. S2d

3. Line 129 should ref Fig. S2g,h but not Fig. S2f,g

4. A statistical test for difference in length between pSTRs and mSTRs would be beneficial (also per motif length)

5. This sentence (starts at line 126) is hard to parse

6. For most pSTRs (the percentage of pSTRs with reference length ≤ 40 bp was 98.2%), mean length differences between called alleles and reference alleles were around zero (Fig. S2e); the genotypes had high quality scores at both the sample level and site level (Fig. S2f, g).

This sentence is a possible repetition of the result above but more understandable? (line 165)

Meanwhile, repeat distances between the most common alleles and reference alleles revealed a symmetric spectrum, with the reference alleles of the vast majority (94.6%) of pSTRs being the most common ones (Fig. 2e; Fig. S6b).

7. What does it mean for a loci to "have at least one repeat away from the reference allele"? It wouldn't be polymorphic if it didn't (line 167)

For dinucleotide STRs, 16.2% (12,559/ 77,617) of loci had at least one repeat away from the reference allele, reflecting the high instability of these pSTRs (Fig. S6b)

8. This sentence needs to be rewritten to be clearer (line 308)

Only one pSTR were found in 5'UTR regions ("chr1:200669896" in DDX59), which was found in high LD with SNP rs6700559 associated with coronary artery disease (Fig. S24b).

9. What is the threshold for a gene to enclose a loci (in the highly variable pSTR within population analysis)?

10. Did the authors control for possible differences between the two subsets in the NyuWa dataset, i.e., the 2,999 samples that were reported (sequenced?) before and the 1,014 newly sequenced samples?

11. A multiplication sign is missing in the p-val threshold in lines 665 and 755
12. Do the diseases associated with SNPs in LD with the pSTR in UBE2L3 have different frequencies within the populations according to the literature? F.ex. It is mostly expanded in East Asian samples, is the frequency of Crohn's disease higher/lower there than in the other populations?
13. Legend for Fig.2 : "The black dashed line indicated the position that x-axis value equal to xx" hard to understand and possibly xx is a number the authors forgot to enter? f is not in bold, like subparts a-e
14. Would be beneficial to see the maximum number of alleles for each motif length, i.e. for each motif length how many alleles does the pSTR with most alleles have?
15. Legend for Fig. S20 says PUDI value, but paper (and literature) says PDI
16. Fig. S19 How are the red points different from the rest? grey are discordant and blue are concordant between the two studies but what are the red dots? Legend says they are found in both studies but then they should be blue or grey?

Reviewer #2 (Remarks to the Author):

Shi et al. describe the properties of 366,034 polymorphic short tandem repeats (pSTRs) in ~4,000 whole genome sequenced Chinese individuals and 2,504 1000 Genomes individuals. They found 3,539 and 1,244 pSTRs associated with gene expression and 3'UTR alternative polyadenylation. While most genetic association studies have focused on individuals of European descent, this is one of the first that thoroughly investigates Han Chinese individuals.

Comments:

I understand that this manuscript (<https://www.biorxiv.org/content/10.1101/2022.08.01.502370v1.full>) was not available yet when the authors wrote and submitted the current manuscript, but the Introduction would greatly benefit expanding the description of GWAS on STRs based on these very recent findings.

The section "pSTR mutational patterns" would benefit from a comparison with previously described STR datasets. It is unclear if the results described in this section are expected or if they include novel observations compared with what is currently known.

In Figure S7, how is the fact that some chromosomes are shorter than 120 Mbp addressed? Also, it would make sense only to consider STRs on a single chromosome arm, rather than the whole chromosome. It would also be interesting to perform the same analysis on the distance from the centromere.

Why were chromatin features only in ESCs used? I understand that the general associations should work independently from the tissue context, but it would be useful to have additional ENCODE samples as a supplemental figure. Figure S12 would be easier to be interpreted if the features in the two heatmaps were sorted in the same order.

The observation that STRs with repeat lengths of 3 and 6 nucleotides are enriched in coding regions is interesting and worth more explanation: these are all in-frame indels that should have lower effects on the transcript and protein function than other frameshift STRs.

I would suggest to remove traits like "educational attainment", "mathematical ability" and "intelligence", as they have a strong environmental and socio-economic component, and might be misinterpreted.

The STR-gene distance threshold in the eSTR analysis should also be expanded to 500kb-1Mb, to be consistent with most eQTL studies. It is also unclear if the threshold used was 10 kb (as described in the Results) or 100 kb (Methods).

Lines 150-152: the statement "we found there were more 151 pSTRs than mSTRs for di-, tri-, and tetranucleotide STRs" should be strengthened by statistical analysis confirming that the differences are not due to chance.

The differences between populations described in Figure 6 would be more evident if the X and Y axis scales were the same across all plots.

Reviewer #1 (Remarks to the Author):

Summary:

The authors present a comprehensive analysis of short tandem repeats (STRs) in a set of 6,487 human genomes which all originate from Asia. STRs are the second most common type of polymorphisms in the human genome, after SNPs and indels. Despite their abundance these variants have not been used extensively in genetic analysis after the advent of short read sequencing. Their underutilization despite their high and valuable information content is mainly due to the algorithmic complexities of analyzing these variants from short read data.

The current manuscript adds to a limited set of publications exploring the patterns of polymorphism of this important variant class and roughly 370 thousand polymorphic STRs (pSTRs) are identified out of the 765 thousand STRs that are analyzed, the rest are monomorphic (mSTRs). The main novelty of the paper is the origin of the sample set since it is the largest analysis to date containing only samples from Asia.

Previously reported patterns for mutational behavior and chromosomal distribution of STRs are replicated both with respect to chromosomal position and functional annotation. Expansions are determined at 113 pSTRs using a previous definition of expansion status, including one pSTR in a 5'UTR region and in LD with a SNP associated with CAD which merits further research. A population analysis of the pSTRs recapitulates previous patterns and separates both super and subpopulations in an expected manner.

An interesting part of the analysis is an allele length distribution comparison between the populations and the analysis of four frequent outlier pSTRs which I feel deserves further analysis and more discussion.

The last two parts of the analysis focus on highly variable pSTRs and known pathogenic pSTRs. The genes enclosing the highly variable pSTRs are shown to be enriched for genes expressed in the cerebral cortex and genes involved in the development and function of the nervous system. For the known pathogenic pSTRs, a small subset of samples alleles equal to or longer than the pathogenicity threshold were observed but no comments are made as to whether the carriers of the alleles suffer from the conditions associated with the expansions.

The following parts of the paper display creative analysis and could be extended and emphasized:

*Expansion analysis (using definition from Press et al.)

*Between population length distribution comparison and analysis of frequent outliers

*Highly variable pSTRs with length distributions that are different between populations

The parts that mostly confirm previously reported results should possibly be moved to a supplement to better enable readers to detect the novel results presented.

The sample set is large but no phenotypes are used for associating the genotypes generated, whether that is due to their unavailability is not addressed and I feel association results would add weight to the paper. Further, despite the fact that the set is larger than previously published set of Asian origin, large parts of the analyses are performed on subsets of the full set or only on samples with origins outside of Asia.

Response: Thanks for your constructive comments. The samples used in this study were not only

from Asia. We jointly analyzed two large cohorts (NyuWa and 1KGP) to obtain a systematic view of the variation of STR in diverse populations, with an emphasis on Han Chinese. The primary goal of this study was to construct a systematic view of STR variation in various populations and to characterize the mutational, functional, and gene-regulatory impacts of polymorphic STRs in the human genome. We also used multiple strategies to nominate pSTRs, such as expansion analysis, length comparison among superpopulations, and identification of highly variable loci within superpopulations. However, as you have noticed, limited by the phenotypic information that we can access, we were unable to determine the contribution of the pSTRs nominated to specific traits or diseases. We hope that our study could serve as a basis for future association studies.

We thank you sincerely for your suggestions. For the expansion analysis, we adopted the metric used by Press et al. to detect modest expansions in the whole call set. During the review process, we found that many loci identified were actually sites with length differences among different superpopulations, as shown in the following figure (expansion loci were colored red).

We thus decided to conduct the expansion analysis within each superpopulation. Through this way we could know whether an expansion locus was population-specific or population-shared. In addition, to improve the coherence of the manuscript, we moved this section “pSTR expansion analysis” behind the section “Highly variable pSTRs”, as “pSTR expansion analysis” was part of the pSTR population analysis and it also related to the final section “STR variation in known disease loci”. The text and figures have been modified. Please find the changes in the revised manuscript.

Thanks for your suggestion. Considering the novelty and significance of section “Highly variable pSTRs”, now we added the figure of this part to the main text (Fig. 7). We also performed more analysis in the section “Highly variable pSTRs”. While the “pSTR lengths comparisons across populations” identified loci with significant changes in mean lengths between different groups, some loci may have different length variability (variance) in different populations. Thus, we used differences of standard deviations and permutation test to identify pSTRs with significant changes in variance of allele lengths among different superpopulations. Among the differential-variability

sites, several loci were associated with GWAS risk SNPs, gene expression regulation, and 3'UTR alternative polyadenylation. The differential variability analysis revealed a different layer of population differentiation signature of pSTR. Please find the updates in the revised manuscript. To accommodate these changes, we have modified the title of two sections: "Population differentiation signatures of pSTR" to "pSTR length comparisons across populations", and "Highly variable pSTRs within superpopulation" to "pSTR length variance analysis". Please find all the updates in the revised manuscript.

Major Comments:

1. Would suggest proof reading of the English in the paper, in some cases this does not affect understanding but in others, sentences are hard to parse.

Response: Thanks for your suggestion. We have tried our best to polish the language in the revised manuscript and marked all modifications in red.

2. It is true that most individuals in the UKB study were white British but the South Asian population was 3063 individuals (only 1000 samples less than the NyuWa set) and the African population was 2976 samples.

Response: In this work, we jointly analyzed two large cohorts (NyuWa and 1KGP) to obtain a systematic view of STR variation in diverse populations, with an emphasis on Han Chinese. The UKB study contained 1,504 Chinese individuals, while 4,103 Chinese individuals were included in our work. We have modified the main text to mention the number of Chinese samples in the UKB: "Moreover, other large-scale studies of pSTRs were also mainly from European ancestry cohorts, and the diversity of pSTRs in East Asia and China is largely undercovered. Even in the UK biobank cohort, there were only 1,504 Chinese samples [38]."

3. Why are there only seven sites (out of 60 known pathogenic) that both GangSTR and Expansion hunter genotype?

Response: ExpansionHunter is a tool for performing targeted searches for a given STR locus (usually dozens of sites), and we collected 60 known pathogenic loci from multiple sources as input for ExpansionHunter. GangSTR is designed to genotype STRs across the genome, and in this study 765,227 autosomal STR loci were used. For the two input variant catalogs, there were only 29 common sites. Of these 29 sites, 22 sites were excluded after strict locus-level filtering for the GangSTR call set. Therefore, only seven sites found in both GangSTR and ExpansionHunter final call sets were used for genotype concordance comparison.

We also checked the genotype concordance of GangSTR and ExpansionHunter for 22 loci that were filtered out in the GangSTR call set (following table), and the median concordance rate of the two tools was 79.6%, lower than that of the seven loci in the final call sets of both tools. This result indicates that our filtering process improved the quality of the GangSTR call set.

Locus ID	Callrate (EH)	AlleleNum (EH)	Callrate (GangSTR)	AlleleNum (GangSTR)	Concordance
----------	---------------	----------------	--------------------	---------------------	-------------

ATXN1	0.7688	25	0.9857	43	0.0061
ATXN10	0.9309	17	0.9975	21	0.9854
ATXN2	0.7629	25	0.8824	29	0.9861
ATXN3	0.9097	31	0.9617	29	0.0012
BEAN1	0.9841	95	0.9372	22	0.0530
C9ORF72	0.7697	21	0.7712	14	0.9611
CACNA1A	0.8351	17	0.9070	17	0.9951
CNBP	0.8478	25	0.9869	30	0.1233
CSTB	0.9470	5	0.8569	10	0.9898
DMPK	0.9824	34	0.9946	35	0.9955
HTT	0.8659	29	0.9112	35	0.8053
JPH3	0.9886	27	0.9986	28	0.9931
MARCHF6	0.9951	30	0.9712	23	0.9139
NOP56	0.9730	14	0.9138	10	0.1630
NUTM2B-AS1	0.9061	16	0.9120	17	0.9924
PPP2R2B	0.9632	27	0.9938	30	0.9968
PRDM12	0.4770	16	0.4728	16	0.0012
RFC1	0.9158	94	0.8326	22	0.7221
STARD7	0.9864	74	0.8626	21	0.7869
TBP	0.5656	27	0.5212	32	0.0850
XYLT1	0.0294	17	0.0046	9	0.0000
YEATS2	0.9047	62	0.8095	19	0.7067

4. This analysis should be repeated per motif length and per major allele length to confirm bias of longer alleles towards contraction and shorter to expansion (line 170)

We observed a higher proportion of alleles longer than the major allele (45.2%) compared to alleles shorter than the major one (31.3%), and the allele frequency monotonically decreased with deviation from the major allele (Fig. 2f; Fig. S6c), coinciding with the stepwise mutation model [55].

Response: Thanks a lot for your suggestion. We repeated this analysis per motif length and per major allele length. Because there were too many situations, we only displayed pSTRs with major allele lengths between 10 and 20 bp, which occupied about 85% of all pSTRs. The following figure shows the results. Consistent trends were observed for both pSTRs with short major allele lengths (e.g., 10 bp) and those with long major allele lengths (e.g., 20 bp). We have added the following figure as a new supplementary figure (new Figure S7), and described it in the main text: “We also performed this analysis per motif length and per major allele length, and similar trends were observed (Fig. S7).”.

5. 0.83% of the polymorphic STRs are shown to have an $R^2 \geq 0.7$ with a SNPs associated with traits or diseases but no further analysis to determine causality was performed. Were these pSTRs located in more functionally important regions than the SNPs they tag f.ex? Or is there some way to do conditional association with the phenotypes that the tagged SNPs are associated with?

Response: Thanks for your advice. We are unable to determine causality or perform a conditional association of STRs because we lack phenotypic information. What we did here was to identify pSTRs in high LD with GWAS SNPs. To determine whether pSTRs and tagged SNPs were located in more functionally important regions, we compared their distance to gene TSS, CpG islands, TAD boundaries, and multiple candidate cis-regulatory elements (cCREs) from the ENCODE project, including promoter-like signature (PLS), proximal enhancer-like signature (pELS), distal enhancer-like signature (dELS), CTCF-only, and DNase-H3K4me3. The following figure shows the results. In general, we found that the GWAS SNPs were significantly closer to TSS, CpG island, PLS, pELS, and dELS than tagged pSTRs (Mann-Whitney U test $p < 0.01$). The result was unsurprising, since GWASs

were based on SNPs, and the SNPs in the GWAS catalog have undergone rigorous screening. In this analysis, we found that 88 pSTRs showed a closer distance to the nine functionally important regions compared to the GWAS SNPs, indicating that these pSTRs potentially contribute more to the phenotype. As STRs were overlooked by most GWASs and other association studies, more research is needed to clarify the contribution of STRs to complex human traits and disorders.

6. A set of 445 samples was used to demonstrate associations of dosages of specific STRs with expression and 3'UTR alternative polyadenylation of genes close by. These samples were however of European and African ancestry, different from the focus of the paper. This analysis seems out of place compared to the other parts of the paper which focuses mostly on characterizing pSTRs in Asian populations. Further, similar analysis has already been published for 311 out of the 445 samples used here, reporting similar results and is cited in the paper.

Response: Thanks for your careful reading. As mentioned above, our study aimed to build a comprehensive resource for STR variation in diverse populations, including five superpopulations from the 1KGP project and the Han Chinese population from the NyuWa dataset. The RNA-seq data were from the Geuvadis consortium and 445 samples were also in the 1KGP cohort. The eSTR and 3'aSTR analysis in this study were used to explore the gene-regulatory effects of pSTRs. Although 311 samples from the RNA-seq dataset were used for the analysis eSTRs in previous study, the associations between pSTRs and the alternative polyadenylation of 3'UTR have not been analyzed. Furthermore, with the high-quality set of pSTRs obtained from the GangSTR, our study extended the previous findings of eSTRs. It would be great to have paired genotypes and transcriptome data available for the NyuWa cohort, but currently the Geuvadis RNA-seq dataset is the only one we can obtain.

7. How is the frequently used 10% FDR chosen? -> This choice should be rationalized and justified either in Methods or Supplement

Response: This threshold (gene-level FDR threshold of 10%) was used in a previous eSTR study by Fotsing et al. (PMID: 31676866), which we have cited in the methods. Since we adopted the eSTR identification pipeline by Fotsing et al. and a comparable sample size was used in our study, we chose this threshold. We have described this in the methods in the original manuscript: "As described before [16], we applied a gene-level FDR threshold of 10% to determine significant STR-gene pairs."

8. I suggest making more use of the trio families, a locus that does not follow Mendelian inheritance could harbor a true microsatellite mutation. Could order violating loci by gq scores and

examine them in decreasing order to deny or confirm the existence of a mutation

Response: Thanks for the suggestion. Because we lacked kinship information for the NyuWa samples, these 58 trio families were inferred with the combination of kinship coefficient and probability of zero identity-by-descent sharing. For each trio, we sorted all STR loci that did not follow Mendelian inheritance (MI) by the GQ score (defined as the minimum value of the GQ scores of the father, mother and child's genotype). The following table shows the results. On average, 16,963 STRs did not follow MI per trio, of which 10,689 loci were highly confident with GQ > 0.9, which may contain real mutations. However, further confirmation involves the identification of *de novo* STR mutations in families, which was not the focus of this work. Several recent studies have identified *de novo* STR mutations (Mitra et al., *Nature*, 2021; Wendt, Pathak, and Polimanti, *medRxiv*, 2022; Steely et al., *bioRxiv*, 2021). We might consider doing a similar analysis in the future if we could obtain more pedigree samples.

sample id of child in a trio	# sites not following mendelian inheritance	# sites with minimum gq score > 0.9	proportion
CN001440	16,085	8,826	0.55
CN001369	19,329	11,678	0.60
CN001311	17,040	9,934	0.58
CN001398	18,440	12,125	0.66
CN004386	20,609	16,322	0.79
CN001387	18,614	12,131	0.65
CN001352	16,489	9,402	0.57
CN001418	18,273	11,048	0.60
CN001376	18,405	11,685	0.63
CN001371	19,096	11,704	0.61
CN001439	12,214	7,018	0.57
CN001344	12,483	7,083	0.57
CN001441	22,576	16,808	0.74
CN004389	21,610	17,062	0.79
CN001422	17,312	10,179	0.59
CN001391	18,671	10,868	0.58
CN001427	14,863	8,578	0.58
CN001308	21,811	15,957	0.73
CN004383	21,426	16,771	0.78
CN001331	15,320	9,023	0.59
CN001431	8,971	5,216	0.58
CN001363	16,964	10,024	0.59
CN001316	19,577	14,070	0.72
CN001399	18,679	11,783	0.63
CN001396	17,059	9,400	0.55
CN001404	12,749	7,219	0.57
CN001370	19,994	15,012	0.75
CN001442	19,952	12,402	0.62
CN001424	15,991	8,788	0.55
CN001321	18,744	12,884	0.69
CN001405	20,697	15,572	0.75
CN001333	16,156	9,842	0.61
CN001324	19,194	12,340	0.64
CN001393	13,185	7,344	0.56
CN001343	21,328	13,660	0.64
CN001384	16,249	11,889	0.73
CN001330	17,922	11,224	0.63
CN001315	17,745	10,053	0.57
CN001432	12,915	7,019	0.54
CN001357	13,519	7,708	0.57
CN001339	14,699	8,922	0.61
CN001337	17,332	9,630	0.56
CN001435	17,011	9,978	0.59
CN001408	19,148	11,576	0.60
CN003149	21,297	13,030	0.61
CN001354	18,091	11,088	0.61
CN001434	15,876	8,752	0.55

CN001430	17,203	10,250	0.60
CN001341	18,068	11,236	0.62
CN004467	19,494	14,831	0.76
CN001413	13,173	7,346	0.56
CN001329	18,360	11,228	0.61
CN001445	18,234	10,964	0.60
CN001443	19,917	11,915	0.60
CN001436	17,774	10,530	0.59
CN001313	17,530	9,925	0.57
CN001390	17,520	10,336	0.59
CN001368	18,796	12,134	0.65
mean	17,548	11,057	0.62

9. Defining STR dosage as the sum of the repeat number of the two alleles is valid but since $10+10=18+2$, I would suggest adding the same analysis with STR dosage defined as the length of the allele further from the ref length or longer allele. Some proteins could have an eSTR with a threshold for which all alleles above or below block expression, this merits thresholding alleles before doing regression. Doing single allele R2 would also work, i.e, creating binary marker for each allele.

Response: We feel sorry that we did not accurately describe the STR dosage used in this study in the original manuscript. The STR dosage was defined as the sum of the deviations in the STR allele repeat numbers from the reference allele, which was similar to previous studies (PMID: 31676866, 26642241). For example, if the reference repeat number for an STR is 2 copies and the two alleles called were 3 copies and 4 copies, the dosage was equal to $(3 - 2) + (4 - 2) = 3$. We have corrected the description in the revised manuscript.

As you suggested, we also defined the STR dosage as the repeat number difference between the longest allele and the reference allele. As the second reviewer suggested, here we set the distance threshold to 500 kb to get STR-gene pairs. Then, we repeated the eSTR identification. Thus, a total of 293,991 gene-STR pairs were tested and 3,278 pairs exhibited significant associations at a gene-level false discovery rate (FDR) of 10%. The following figure shows the results. The associations were corroborated by permutation controls (a) and covered all the examples shown previously in (b). The Venn plot shows the comparison of significantly associated eSTR-gene pairs based on the two different dosage definitions (c). The set based on the sum of the deviations in the STR allele repeat numbers from the reference allele included more eSTRs and covered 70% (2,314 of 3,278) eSTRs in the set based on the longer allele. As shown in the figure below (b), the diversity of STR dosage between samples was reduced compared to the previous definition. For example, genotype 0/1 and 1/1 have the same dosage 0 or 1. Therefore, we kept the original definition of STR dosage and only changed the distance threshold in the revised manuscript.

The following figure shows the new Figure 4. We have also updated all the corresponding supplementary figures and text in the sections “pSTR gene-regulatory effects” and “Materials and Methods”.

10. Why use the dosage effect (sum of two alleles) in length comparison between populations? I suggest weighing each allele length with its frequency in the population to obtain a more accurate value for average STR length per population.

Response: In identifying STRs with significant length differences between superpopulations, we summed the allele repeat numbers of each locus to represent the length of the STR in a given sample. This definition was similar to a previous study that detected length differences of VNTRs in superpopulations (PMID: 34244228), which we have cited in our manuscript. In this way, for each locus, we got a single number for each sample and a length distribution for each group of samples. Then we could use fold change and statistical test to measure the length differences between different population groups.

We also tried the way you suggested. We calculated the average STR length per population as the sum of the product of each allele length and its frequency. Then, we calculated the difference in the average length of each locus between any two superpopulations, and the top 1000 loci with the largest difference were considered as population differential loci. We then compared the loci identified by this method with the loci identified before and found that almost all of the loci identified here were in the loci identified before. The number of overlapping loci among superpopulations is as follows: AFR vs. AMR, 1000; AFR vs. EUR, 999; AMR vs. EAS, 994; AMR vs. SAS, 981; EUR vs. EAS, 998; AFR vs. EAS, 1000; AFR vs. SAS, 1000; AMR vs. EUR, 967; EAS vs. SAS, 995; EUR vs. SAS, 989. Based on this observation, we kept the original definition of STR length in this analysis.

Minor comments

1. It needs to be clearer that both Beyter et al. and Mukamel et al. cited in the introduction cover only VNTRs (motifs ≥ 7 bp), not STRs.

Response: Thanks for your careful reading. We have replaced these two studies with two recent preprints that covered STRs: "Recently, a phenome-wide association study identified 426 tandem repeat-phenotype associations, in which GT repeats in *NCOA6* and 'ease of skin tanning' were the most significant [29]. Furthermore, Margoliash et al. tested the association between STRs and blood and serum traits using imputed STR genotypes and identified 118 high-confidence STR-trait associations [30]."

2. Line 126 should reference Fig. S2e but not Fig. S2d

Response: We have corrected this.

3. Line 129 should ref Fig. S2g,h but not Fig. S2f,g

Response: We have corrected this.

4. A statistical test for difference in length between pSTRs and mSTRs would be beneficial (also per motif length)

Response: Thanks for the suggestion. We performed the Mann-Whitney U test for the difference in locus length (represented by the repeat number of the reference allele) between pSTR and mSTR

per motif length. The following figure shows the results. For each motif length, pSTRs were significantly longer than mSTRs (Mann-Whitney U test, $P < 0.01$). We have added this to the manuscript: "In addition, the reference allele lengths of pSTRs were significantly longer than those of mSTRs (Fig. S5c; median of pSTRs: 15, median of mSTRs: 12).".

5. This sentence (starts at line 126) is hard to parse

Response: Thanks for your careful reading. We have corrected this. The text now reads: "For pSTRs with reference allele lengths ≤ 40 bp, which accounted for 98.2% of all pSTRs, the mean differences in length of each allele called compared to the reference allele were approximately zero (Fig. S2f);".

6. For most pSTRs (the percentage of pSTRs with reference length ≤ 40 bp was 98.2%), mean length differences between called alleles and reference alleles were around zero (Fig. S2e); the genotypes had high quality scores at both the sample level and site level (Fig. S2f, g).

This sentence is a possible repetition of the result above but more understandable? (line 165)

Meanwhile, repeat distances between the most common alleles and reference alleles revealed a symmetric spectrum, with the reference alleles of the vast majority (94.6%) of pSTRs being the most common ones (Fig. 2e; Fig. S6b).

Response: These two were different analyses with different purposes. The former ("mean length differences between called alleles and reference alleles") was to check if there was a strong bias toward calling deletions vs. insertions compared to the reference, which could indicate a problem. The latter was to compare the most common alleles with the reference alleles, which could reflect how well the most common alleles are represented in the human reference genome.

7. What does it mean for a loci to "have at least one repeat away from the reference allele"? It wouldn't be polymorphic if it didn't (line 167)

For dinucleotide STRs, 16.2% (12,559/ 77,617) of loci had at least one repeat away from the reference allele, reflecting the high instability of these pSTRs (Fig. S6b)

Response: Thanks for your careful reading. We have corrected this: "For dinucleotide STRs, the major alleles of 16.2% (12,559/77,617) of the loci had at least one repeat away from the reference alleles, reflecting the high instability of these pSTRs (Fig. S6b).".

8. This sentence needs to be rewritten to be clearer (line 308)

Only one pSTR were found in 5'UTR regions ("chr1:200669896" in *DDX59*), which was found in high LD with SNP rs6700559 associated with coronary artery disease (Fig. S24b).

Response: We have corrected this. The text now reads: "Notably, the pSTR "chr1:200669896" in the 5'UTR of *DDX59* was expanded in the NyuWa dataset and East Asians in the 1KGP and was in high LD with SNP rs6700559, which is associated with coronary artery disease (Fig. S30b).".

9. What is the threshold for a gene to enclose a loci (in the highly variable pSTR within population analysis)?

Response: As described in the "Materials and Methods -- Functional annotation", we used Variant Effect Predictor (VEP) to annotate pSTRs, with parameters '--pick --sift b --polyphen b --hgvs --symbol --canonical --biotype --protein --domains --uniprot --tsl --numbers --distance 2000,1000". For each pSTR, we just used the gene assigned by VEP as its host gene. Therefore, the distance threshold was upstream 2 kb or downstream 1 kb of a gene.

10. Did the authors control for possible differences between the two subsets in the NyuWa dataset, i.e., the 2,999 samples that were reported (sequenced?) before and the 1,014 newly sequenced samples?

Response: During the review process, we performed PCA and found no systematic differences between the two subsets of NyuWa samples.

11. A multiplication sign is missing in the p-val threshold in lines 665 and 755

Response: We have corrected this.

12. Do the diseases associated with SNPs in LD with the pSTR in *UBE2L3* have different frequencies within the populations according to the literature? F.ex. It is mostly expanded in East Asian samples, is the frequency of Crohn's disease higher/lower there than in the other populations?

Response: This is an interesting question. We identified four Crohn's disease-associated SNPs in high LD with the pSTR in *UBE2L3*, and the following table shows their allele frequencies (AF) (data from gnomAD v3.1.2). We excluded AMR here because it is a heterogeneous superpopulation. All of these SNPs show the highest AF in EAS, followed by SAS, where the AF of AFR is the lowest. This trend is highly consistent with the repeat length distribution of this pSTR in different

superpopulations (Fig. S27a). For the disease burden of Crohn's disease, the reported incidence values in North America, Oceania, and Europe were higher than those in Asia and Africa, and Africa had the lowest reported incidence values (PMID: 29050646, 31648971). The Crohn's disease is a complex disease affected by both genetic and environmental factors, and the incidence values reported were complicated by the level of attention paid by different countries. The potential association between pSTRs and Crohn's disease needs further research.

SuperPop	rs2266959	rs2256609	rs2266961	rs181359
EAS	0.409	0.4132	0.4132	0.507
SAS	0.2663	0.2663	0.2658	0.427
EUR	0.1809	0.1818	0.1809	0.1925
AFR	0.01978	0.01978	0.03165	0.1068

13. Legend for Fig.2 : "The black dashed line indicated the position that x-axis value equal to xx" hard to understand and possibly xx is a number the authors forgot to enter? f is not in bold, like subparts a-e

Response: This is the number that we forgot to enter. We have corrected this: "The black dashed line indicates the mean value (4.26) of allele numbers of pSTRs."

14. Would be beneficial to see the maximum number of alleles for each motif length, i.e. for each motif length how many alleles does the pSTR with most alleles have?

Response: Thanks for your suggestion. We have checked the maximum number of alleles for each motif length. The maximum allele numbers of di-, tri-, tetra-, penta-, and hexanucleotide pSTR were 51, 38, 73, 105 and 15, respectively. We have added this to the revised manuscript.

15. Legend for Fig. S20 says PUDI value, but paper (and literature) says PDUI

Response: The correct spelling is PDUI, and we have corrected it.

16. Fig. S19 How are the red points different from the rest? grey are discordant and blue are concordant between the two studies but what are the red dots? Legend says they are found in both studies but then they should be blue or grey?

Response: Sorry, we did not express it clearly. The text now reads: "The blue points indicate eSTRs whose directions of effect were concordant in two studies, and the gray points denote eSTRs with discordant directions of effect for that eSTR. The eSTRs detected in both studies are colored red, regardless of the concordance of effect." We also modified the caption of Figure 4b.

Reviewer #2 (Remarks to the Author):

Shi et al. describe the properties of 366,034 polymorphic short tandem repeats (pSTRs) in ~4,000 whole genome sequenced Chinese individuals and 2,504 1000 Genomes individuals. They found 3,539 and 1,244 pSTRs associated with gene expression and 3'UTR alternative polyadenylation. While most genetic association studies have focused on individuals of European descent, this is one of the first that thoroughly investigates Han Chinese individuals.

Response: Thanks for your helpful comments.

Comments:

I understand that this manuscript (<https://www.biorxiv.org/content/10.1101/2022.08.01.502370v1.full>) was not available yet when the authors wrote and submitted the current manuscript, but the Introduction would greatly benefit expanding the description of GWAS on STRs based on these very recent findings.

Response: Thanks for your suggestion. We have now changed the text of the introduction to describe this article: “Recently, a phenome-wide association study identified 426 tandem repeat-phenotype associations, in which GT repeats in *NCOA6* and ‘ease of skin tanning’ were the most significant [29]. Furthermore, Margoliash et al. tested the association between STRs and blood and serum traits using imputed STR genotypes and identified 118 high-confidence STR-trait associations [30].”.

The section “pSTR mutational patterns” would benefit from a comparison with previously described STR datasets. It is unclear if the results described in this section are expected or if they include novel observations compared with what is currently known.

Response: Thanks for your suggestion. We have modified the text in this section and added comparisons with previous studies where appropriate. In summary, observations shown in Figure 2 were largely confirmatory, but our study expanded the number, quality, and especially the diversity of genomes analyzed for STRs. For the enrichment of pSTRs in telomeres, our finding that only hexameric pSTRs enriched in subtelomeres extended previous reports, since previous studies mainly focused on tandem repeats and VNTRs and did not discriminate STRs with different motif lengths. Regarding the correlations between STRs and chromatin features, our analysis was novel. Although associations between STR variability and epigenetic regulation have been reported in several studies (reviewed in PMID: 29398703), to our knowledge, no systematic comparison between STR variation and multiple epigenetic features has been made. Furthermore, owing to the high-quality mSTR call set, we compared pSTRs with mSTRs in many analyzes to better understand the mutational properties of STRs.

In Figure S7, how is the fact that some chromosomes are shorter than 120 Mbp addressed? Also, it would make sense only to consider STRs on a single chromosome arm, rather than the whole chromosome. It would also be interesting to perform the same analysis on the distance from the centromere.

Response: Thanks for your reminding. We didn't notice this problem before. So, the STR count for each bin was calculated for all chromosomes that still exist along the x-axis coordinates. Also, we are sorry that we drew Figure S7 using the total count of STRs as the y-axis. In the revised manuscript, we repeated this analysis by using the average count of STRs as the y-axis and excluded bins that have less than 5 chromosomes. The result is shown in the new Figure S8. Furthermore, we performed this analysis for the p arms (new Fig. S8a) and the q arms (new Fig. S8b), respectively, with the exclusion of 5 p arms of acrocentric chromosomes (13-15,21,22). Similar to our previous findings, hex-nucleotide pSTRs showed significant enrichment in the telomere regions for both the p arms (1.6-fold increase) and the q arms (1.8-fold increase). We have modified the text in the “pSTR mutational patterns” and “Method - Subtelomeric enrichment of STRs” sections. The text

now reads: “Compared with other regions on the chromosome, there was a significant increase (1.6-fold increase for the p arms and 1.8-fold increase for the q arms) in hexameric pSTR density in the last 5 Mbp of chromosome arms (Permutation P < 1x10⁻⁴).”.

We also performed a similar analysis on the distance from the centromere and the result is shown in the following figure (p arms (a) and q arms (b)). Both pSTRs and mSTRs were underrepresented around the centromere regions. In the GangSTR reference file that we used, only 0.05% (418 in 784,553) of the loci were located in the ± 5 kb region of the centromere region. This is partly due to the poor assembly of the GRCh38 reference genome around the centromere regions.

Why were chromatin features only in ESCs used? I understand that the general associations should work independently from the tissue context, but it would be useful to have additional ENCODE samples as a supplemental figure. Figure S12 would be easier to be interpreted if the features in the two heatmaps were sorted in the same order.

Response: Thanks for your suggestion. We used ESC data because they would show a profile closer to germline cells. Since this study observed germline STRs, not somatic ones, seeing correlations with germline cells, such as oocytes, sperms, cells during spermatogenesis and oogenesis, would be ideal. But public epigenetic data of those cells are not as common as ESCs. That was why we

used chromatin features from H1-ESCs, whose epigenetic data was available from public sources such as the ENCODE project. To supplement our findings in H1-ESCs, we performed a similar analysis using H9-ESCs, and similar correlations were observed. The following figure shows the results, and we have added it to supplement Figure S13. We also modified the text in the sections “pSTR mutational patterns” and “Method - Correlation analysis of STR and genomic features” to describe this.

As you suggested, we also sorted the features in heatmaps in the same order.

The observation that STRs with repeat lengths of 3 and 6 nucleotides are enriched in coding regions is interesting and worth more explanation: these are all in-frame indels that should have lower effects on the transcript and protein function than other frameshift STRs.

Response: Thanks for your suggestion. We have modified the text: “In contrast, tri- and hexameric STRs were overrepresented in 5’UTRs and CDSs compared with other STR types. This was likely because triplet STRs are all in-frame indels that should have less of an effect on transcript and

protein function than other nontriplet STRs [6,68].”.

I would suggest to remove traits like “educational attainment”, “mathematical ability” and “intelligence”, as they have a strong environmental and socio-economic component, and might be misinterpreted.

Response: Thanks for your advice. We have deleted traits “self reported educational attainment”, “mathematical ability”, and “intelligence” in this analysis, and then updated the Figure S16 in the original manuscript and modified the main text: “We identified 2,871 pSTRs that were in high LD (‘tagged’) with at least one GWAS SNP ($r^2 \geq 0.7$), accounting for approximately 0.78% of all pSTRs (Fig. S17a; Table S6). The major allele frequencies for 303 of these pSTRs were less than 0.5. The traits with the highest numbers of GWAS-tagged pSTRs included height, body mass index, and heel bone mineral density (Fig. S17b).”.

The STR-gene distance threshold in the eSTR analysis should also be expanded to 500kb-1Mb, to be consistent with most eQTL studies. It is also unclear if the threshold used was 10 kb (as described in the Results) or 100 kb (Methods).

Response: Sorry, we did not describe the threshold correctly. We used the 100 kb threshold in our analysis according to previous eSTR studies (PMID: 31676866, 26642241). Thank you for your suggestion. We have changed the threshold to 500 kb and performed this analysis again. We have modified the main text and the corresponding figures. The text now reads: “For eSTR associations, a total of 293,991 gene-STR pairs were tested, and 4,131 pairs (4,131 genes, 3,273 pSTRs) exhibited significant associations at a gene-level false discovery rate (FDR) of 10% (Table S7; Fig. S19)...Using a similar method, we tested 967,309 transcript-STR pairs to identify 3’aSTRs, and 1,984 pairs (1,984 transcripts, 1,117 pSTRs) were significant at transcript-level FDR < 10% (Fig. 4c; Table S8).”

Lines 150-152: the statement “we found there were more pSTRs than mSTRs for di-, tri-, and tetranucleotide STRs” should be strengthened by statistical analysis confirming that the differences are not due to chance.

Response: Thanks for your advice. We performed the chi-square test for di-, tri-, and tetranucleotide STRs. Significant differences between observation and expectation (the number of

pSTRs was equal to the number of mSTRs) were observed for each motif length. We also modified the text, and the sentence now reads: “we found that there were more pSTRs than mSTRs for di- ($\chi^2 P < 1 \times 10^{-22}$), tri- ($\chi^2 P < 1 \times 10^{-22}$), and tetranucleotide ($\chi^2 P = 8 \times 10^{-22}$) STRs.”

The differences between populations described in Figure 6 would be more evident if the X and Y axis scales were the same across all plots.

Response: Thanks for your advice. We have drawn this figure in the way as you suggested and we have replaced the original Figure 6. Also, log2 fold changes in Figure 6 and original Figure S28 did not meet the general reading habits. For example, the previous first subplot showed the log2 fold changes of SAS vs. AFR. Now we have updated the two figures to make it more readable. For example, the first subplot shows the log2 fold changes of AFR vs. SAS.

REVIEWERS' COMMENTS

Reviewer #1 (Remarks to the Author):

The authors have largely addressed the concerns presented in our previous review but a few minor issues remain or were detected during this second revision round.

Line 156: Increased mutation in shorter motifs has been shown previously and should have a consistent with previous results statement and citation to those papers.

Line 158: Increased mutation rate in longer reference allele lengths has been shown previously and should have a consistent with previous results statement and citation to those papers.

Line 260: If the trait being listed refers to a bone in a person's foot, the correct spelling is heel bone density rather than heal bone density.

Line 295: The text here does not match the image referenced. The biggest effect for 3'aSTRs is in noncoding exons according to Fig. S23.

Line 386: If possible, add definition of the function of DHTKD1 and MGAT5 like for the other genes mentioned.

Fig S29: Caption is missing definition of the grey horizontal line like the one given for the gray vertical line

Line 433: If GNA12 has a known function, it should be added to the text.

Line 500: pSTRs in CDSs tended to be less variable, and their host genes were under stronger selection. (Are the genes harbouring pSTRs really under stronger selection?)
line 239-243 says they have higher tolerance to inactivation and should be under weaker selection: From the perspective of pSTR-containing genes, we utilized another measure: the loss-of-function observed/expected upper bound fraction (LOEUF), where higher LOEUF scores indicate higher tolerances to inactivation for given genes [70]. Higher LOEUF scores were observed for genes with pSTRs in coding regions than those with pSTRs in intronic regions (Fig. 3f).

Reviewer #2 (Remarks to the Author):

The Authors are addressed all my concerns.

Reviewer #1 (Remarks to the Author):

The authors have largely addressed the concerns presented in our previous review but a few minor issues remain or were detected during this second revision round.

Response: Thank you very much for your time and helpful comments.

Line 156: Increased mutation in shorter motifs has been shown previously and should have a consistent with previous results statement and citation to those papers.

Response: We have modified this sentence. The text now reads: "For the reference STR set we analyzed, there were more pSTRs than mSTRs for di- ($\chi^2 P < 1 \times 10^{-22}$), tri- ($\chi^2 P < 1 \times 10^{-22}$), and tetranucleotide ($\chi^2 P = 8 \times 10^{-22}$) STRs (Fig. S5a, b), implying that STRs with shorter repeat units had higher plasticity, as previously reported [33,54]."

Line 158: Increased mutation rate in longer reference allele lengths has been shown previously and should have a consistent with previous results statement and citation to those papers.

Response: We have modified this sentence. The text now reads: "In addition, the reference allele lengths of the pSTRs were significantly longer than those of the mSTRs (Fig. S5c; median of pSTRs: 15, median of mSTRs: 12), suggesting an increased mutation rate in longer reference allele lengths [3,33]."

Line 260: If the trait being listed refers to a bone in a person's foot, the correct spelling is heel bone density rather than heal bone density.

Response: Thank you for your careful reading. We have corrected this: "Traits with the highest numbers of GWAS-tagged pSTRs included height, body mass index, and mineral density of the heel bone (Fig. S17b)."

Line 295: The text here does not match the image referenced. The biggest effect for 3'aSTRs is in noncoding exons according to Fig. S23.

Response: We have corrected this: "Concordantly, eSTRs in 5'UTRs and 3'aSTRs in noncoding exons and 3'UTRs had larger effect sizes than those in other regions (Fig. S23)."

Line 386: If possible, add definition of the function of DHTKD1 and MGAT5 like for the other genes mentioned.

Response: Thank you for your suggestions. For the *DHTKD1* gene, we added a sentence to describe its function: "The *DHTKD1* gene encodes a dehydrogenase involved in mitochondrial energy production [82] and mutations in this gene have been associated with 2-aminoadipic 2-oxoadipic aciduria and Charcot-Marie-Tooth Disease Type 2Q [83,84]."

For the *MGAT5* gene, we modified the sentence to describe its function: "The pSTR in *MGAT5* gene, encoding an important enzyme involved in the synthesis of glycoprotein oligosaccharides, had longer repeats in individuals in Africa, and it showed a similar bimodal distribution in East Asians as the pSTR in *MYPN* (Fig. S27d)."

Fig S29: Caption is missing definition of the grey horizontal line like the one given for the gray vertical line

Response: We have added a description for the gray horizontal lines in this caption: “a Pairwise comparisons of pSTR length variance between superpopulations from the 1KGP are shown using volcano plots. The gray horizontal lines indicate the q-value cutoff $-\log_{10}(0.01)$ for pSTRs with differential length variances.”.

Line 433: If GNA12 has a known function, it should be added to the text.

Response: We have added a sentence to describe the function of *GNA12*: “The *GNA12* gene encodes a subunit of the G proteins and functions as modulators or transducers in various transmembrane signaling systems [87,88].”.

We also added a short description of the gene *PREPL*: “...could regulate the 3’UTR alternative polyadenylation of multiple transcripts of *PREPL*, a member of the prolyl oligopeptidase subfamily of serine peptidases.”.

Line 500: pSTRs in CDSs tended to be less variable, and their host genes were under stronger selection. (Are the genes harbouring pSTRs really under stronger selection?)

line 239-243 says they have higher tolerance to inactivation and should be under weaker selection:

From the perspective of pSTR-containing genes, we utilized another measure: the loss-of-function observed/expected upper bound fraction (LOEUF), where higher LOEUF scores indicate higher tolerances to inactivation for given genes [70]. Higher LOEUF scores were observed for genes with pSTRs in coding regions than those with pSTRs in intronic regions (Fig. 3f).

Response: Thank you for your careful reading. The sentence in Line 500 was kind of misleading. Using the LOEUF metric, we found that protein-coding genes containing pSTRs in their CDSs had higher tolerance to inactivation than genes with pSTRs in their intronic regions, indicating that pSTRs which directly impact CDSs were under strong selective constraint. This analysis compares CDSs and coding introns, but not genes under strong selection versus genes under weak selection. We have corrected this. The text now reads: “We found that pSTRs in CDSs tended to be less variable and were under strong selective constraint.”.

Reviewer #2 (Remarks to the Author):

The Authors are addressed all my concerns.

Response: Thank you very much for your time and effort in reviewing our manuscript.